# A broad mutational target explains a fast rate of phenotypic evolution

**Fabrice Besnard[1,2†]\*, Joao Picao-Osorio[1†], Clément Dubois[1], Marie-Anne Félix[1]\***

[1]Institut de Biologie de l'École Normale Supérieure, CNRS, Inserm, Paris, France; [2]Laboratoire Reproduction et Développement des Plantes, Univ Lyon, ENS de Lyon, UCB Lyon 1, CNRS, INRAE, Inria, Lyon, France

**Abstract** The rapid evolution of a trait in a clade of organisms can be explained by the sustained action of natural selection or by a high mutational variance, that is the propensity to change under spontaneous mutation. The causes for a high mutational variance are still elusive. In some cases, fast evolution depends on the high mutation rate of one or few loci with short tandem repeats. Here, we report on the fastest evolving cell fate among vulva precursor cells in *Caenorhabditis* nematodes, that of P3.p. We identify and validate causal mutations underlying P3.p's high mutational variance. We find that these positions do not present any characteristics of a high mutation rate, are scattered across the genome and the corresponding genes belong to distinct biological pathways. Our data indicate that a broad mutational target size is the cause of the high mutational variance and of the corresponding fast phenotypic evolutionary rate.

**\*For correspondence:**
fabrice.besnard@ens-lyon.fr (FB);
felix@biologie.ens.fr (M-AF)

[†]These authors contributed equally to this work

**Competing interests:** The authors declare that no competing interests exist.

## Introduction

In a given phylogenetic clade of organisms, some phenotypic traits evolve faster than others or faster than in other groups. When they in addition appear to evolve directionally, this is called an evolutionary trend (*Gould, 1988*; *McShea, 1994*; *McShea, 2000*). Classical examples are the reduction in digit number of horses, the increase in brain size in hominids or the change in fractal complexity of suture lines in the fossil record of ammonites (*McNamara, 2006*). A possible explanation for fast evolutionary change of a trait is the sustained action of natural selection on the trait, acting in either a directional or a diversifying manner. A second explanation arises from the fact that the available phenotypic variation onto which natural selection acts is not uniform along all axes of phenotypic space (developmental constraints or the 'arrival of the fittest') (*Gould, 1977*; *Cheverud, 1984*; *Alberch and Gale, 1985*; *Maynard Smith et al., 1985*; *Arthur, 2004*; *Dichtel-Danjoy and Félix, 2004*; *Denver et al., 2005*; *Rifkin et al., 2005*; *Landry et al., 2007*; *Stoltzfus and Yampolsky, 2009*; *Wagner, 2014*; *Hether and Hohenlohe, 2014*; *McGuigan and Aw, 2017*; *Hine et al., 2018*). Indeed, upon random mutation, some axes of phenotype space are more readily explored than others. In other terms, the mutational variance may not be equal along different axes of phenotype space and this may sufficiently affect the rate of evolution at the phenotypic level. Natural selection may act in an orthogonal manner to the mutational variance in phenotype space (that is, may select on a trait with low mutational variance); and along the axis of high mutational variance, it may act in the same direction or in the opposite direction. Phenotypic evolution then results from the combination of the mutational variance and natural selection.

The present study addresses the causes of high mutational variance along some directions of phenotypic space. Two non-mutually exclusive explanations may underlie such phenomenon, the first at the molecular level, the second at the level of genotype-phenotype mapping: (1) some DNA sequences, such as short tandem repeats, are more prone to spontaneous mutation; (2) a higher mutational variance could be due to a higher mutational target size affecting this phenotype. These two factors may act jointly.

**eLife digest** Heritable characteristics or traits of a group of organisms, for example the large brain size of primates or the hooves of a horse, are determined by genes, the environment, and by the interactions between them. Traits can change over time and generations when enough mutations in these genes have spread in a species to result in visible differences.

However, some traits, such as the large brain of primates, evolve faster than others, but why this is the case has been unclear. It could be that a few specific genes important for that trait in question mutate at a high rate, or, that many genes affect the trait, creating a lot of variation for natural selection to choose from.

Here, Besnard, Picao-Osorio et al. studied the roundworm *Caenorhabditis elegans* to better understand the causes underlying the different rates of trait evolution. These worms have a short life cycle and evolve quickly over many generations, making them an ideal candidate for studying mutation rates in different traits.

Previous studies have shown that one of *C. elegans'* six cells of the reproductive system evolves faster than the others. To investigate this further, Besnard, Picao-Osorio et al. analysed the genetic mutations driving change in this cell in 250 worm generations. The results showed that five mutations in five different genes – all responsible for different processes in the cells – were behind the supercharged evolution of this particular cell. This suggests that fast evolution results from natural selection acting upon a collection of genes, rather than one gene, and that many genes and pathways shape this trait.

In conclusion, these results demonstrate that how traits are coded at the molecular level, in one gene or many, can influence the rate at which they evolve.

In the first case, mutational hotspots affecting the phenotype disproportionately increase mutational variance for this trait. Specifically, short repeat regions in a gene may favour DNA replication slippage and recombination, leading to gain or loss of repeats (*Heale and Petes, 1995*; *Gemayel et al., 2010*), or result in fragile DNA conformation susceptible to double-strand breaks (*Xie et al., 2019*). Such highly mutable repeats may lie in a coding region (*Verstrepen et al., 2005*) or within regulatory sequences of a gene (*Vinces et al., 2009*; *Chan et al., 2010*). Their variation has been shown to affect various phenotypes in different organisms (*Levdansky et al., 2007*; *Undurraga et al., 2012*; *Gemayel et al., 2017*; *Dai and Holland, 2019*) and in humans to lead to diseases such as Huntington and fragile X syndromes (*Budworth and McMurray, 2013*). Consequently, the high mutability of some DNA regions may accelerate the evolution of specific traits. Examples are the fast-evolving dog head shape (*Fondon and Garner, 2004*) or the recurrent pelvic fin reduction in sticklebacks (*Chan et al., 2010*; *Xie et al., 2019*).

In the second case, the higher mutational variance of a phenotype may be due to a larger mutational target size rather than a high mutation rate at a given locus: the mutational variance increases with the number of genes (and size of gene regions) whose mutation alters the phenotype. This may be the case for a phenotype that is sensitive to small quantitative alterations, for example in biochemical pathways. The construction of such a trait may indeed be affected by mutations at many loci, many of which may only affect the trait at low penetrance. In another case, bacterial tolerance to antibiotics, mutations to tolerance are frequent because mutations affecting bacterial growth or lag time result in tolerance (*Girgis et al., 2012*; *Girgis et al., 2009*; *Fridman et al., 2014*; *Brauner et al., 2016*; *Khare and Tavazoie, 2020*). Some traits are indeed known to be highly polygenic in natural populations. Some authors even proposed an 'omnigenic' model, where phenotypic variation may result from variation at many genes outside the core pathways known to regulate the phenotype (*Boyle et al., 2017*). This model fits quantitative genetic data of human diseases (*Liu et al., 2019*). However, the number of loci segregating in natural populations also depends on factors such as population structure and selection. To address the origin of a high mutational variance, a more direct approach is needed and more data need to be collected to evaluate how much and in which context each of the above scenarios - highly mutable loci versus a broad mutational target - contributes to a fast rate of phenotypic evolution.

We use the nematode vulva to explore this question. This developmental system relies on six precursor cells, with several useful features: (1) the developmental fate of the six homologous cells can be followed in a wide range of species; (2) the mutational variance of the different precursor cells can be compared on the same scale; (3) much knowledge has been accumulated on the specification of vulval precursor cell (VPC) fates through laser cell ablation studies and developmental genetics. The six vulva precursors are born aligned along the ventral epidermis of the young larvae and are numbered P3.p to P8.p from anterior to posterior (*Figure 1a*). The six cells initially share an identical fate of ventral epidermal blast cells. Under the influence of several signalling pathways, each precursor cell differentiates with a specific terminal fate, creating reproducible patterns of cell fates shared by taxonomic groups of varying size (*Figure 1*). As showed earlier, the developmental fate of one of these six cells, P3.p, by far evolves faster than that of the other Pn.p cells, both within and among species in the *Caenorhabditis* genus (*Delattre and Félix, 2001*; *Kiontke et al., 2007*; *Braendle et al., 2010*; *Pénigault and Félix, 2011a*). While P5.p, P6.p and P7.p divide several times to form the vulva under the influence of EGF and Notch signaling, P4.p and P8.p most often divide once and their daughters fuse with the large epidermal syncytium hyp7 at the end of the third larval stage (L3). Their fate does not evolve in most of the *Caenorhabditis* genus. In contrast, P3.p may

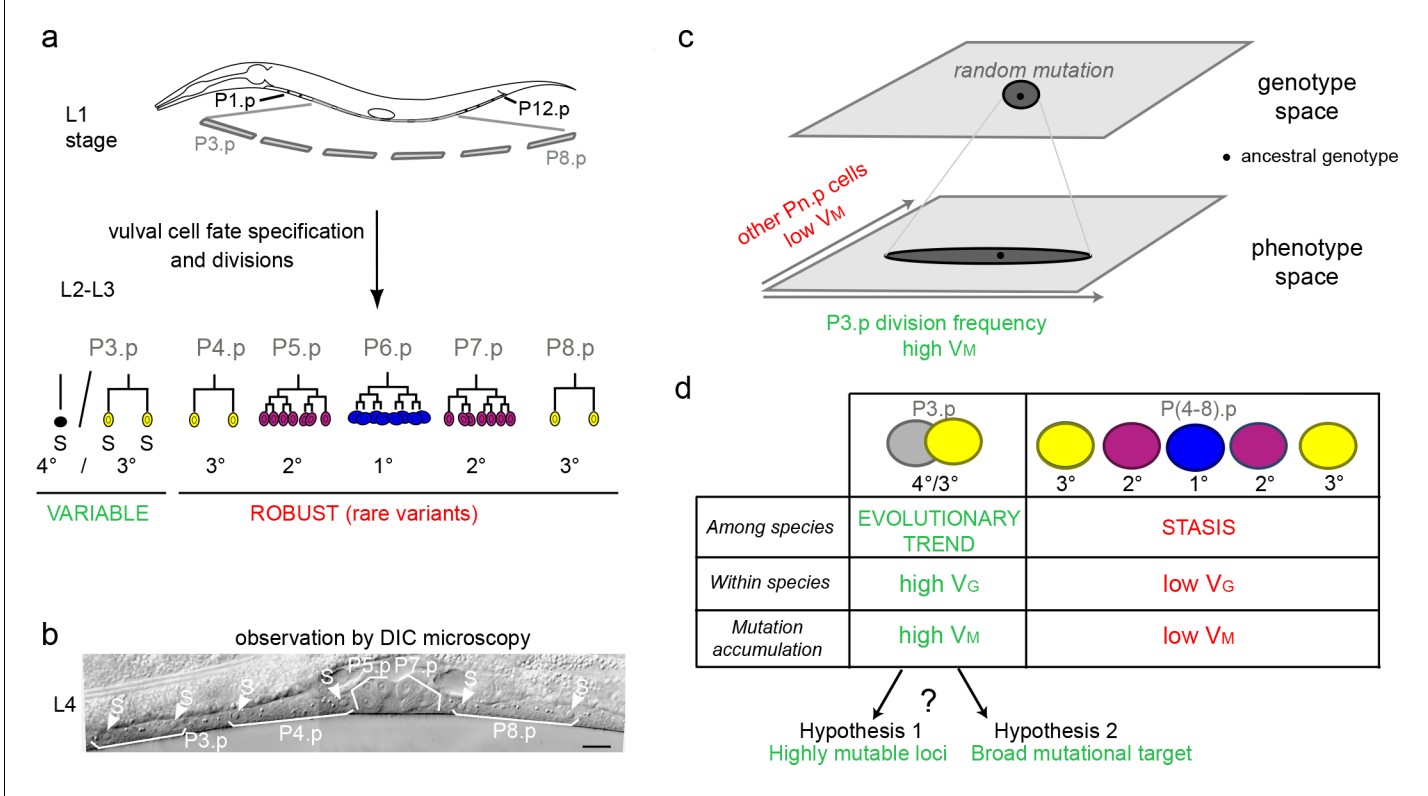

**Figure 1.** Specific evolutionary features of P3.p among vulva precursor cells and the question of the origin of its high mutational variance. (**a**) Schematic description of development of the six vulva precursor cells (VPCs). The six cells P(3-8).p are born during the L1 larval stage. At the end of larval stage L2, P3.p either fuses with the surrounding hypodermal syncytium (hyp7) or escapes fusion like the other VPCs. The VPCs that have not fused divide in the L3 stage according to a fixed fate and lineage (1°, 2° and 3° fates, color-coded). (**b**) Nomarski picture of a mid-L4 stage animal showing the descendants of VPCs. In this individual, P3.p divides like P4.p and P8.p, as shown by the presence of two nuclei per mother Pn.p cell (labeled 'S' for syncytial). (**c**) Schematic genotype-phenotype map for the Pn.p cells, showing that P3.p has a high mutational variance. The black dot depicts the ancestral genotype and phenotype, and the dark grey shape schematizes the distribution after random mutation. (**d**) Unlike P(4-8).p, P3.p displays evolutionary change among *Caenorhabditis* species (evolutionary trend), a high polymorphism within species (standing genetic variance $V_G$), and a high mutational variance ($V_M$) found in mutation accumulation lines (*Delattre and Félix, 2001*; *Braendle et al., 2010*; *Pénigault and Félix, 2011a*). The high mutational variance of P3.p may be explained by a high mutation rate at specific loci or by a broad mutational target.

The online version of this article includes the following figure supplement(s) for figure 1:

**Figure supplement 1.** Comparison of mutational and standing genetic variance among different vulva precursor cells.

either fuse to the hyp7 syncytium already at the end of the L2 stage (with no further cell division possible) or divide once in the L3 stage (*Sternberg, 2005*; *Félix, 2012*). For simplicity, we will refer to this trait as a binary choice between absence or presence of division, which we quantify as a frequency of division in an isogenic population. Isogenicity of the population is obtained easily in the two nematode species we use here, *C. elegans* and *C. briggsae,* because they reproduce through selfing (with the possibility of controlled outcrossing with males for genetic analysis).

We previously showed using mutation accumulation (MA) lines that the particularly fast rate of phenotypic evolution of P3.p fate in the *Caenorhabditis* genus is very likely explained by its high mutational variance (*Braendle et al., 2010*). MA experiments are ideal to test whether some traits vary more than others upon spontaneous mutation and to address the origin of variation in mutational variance. Since the effect of selection is reduced to a minimal fertility requirement at each random generational bottleneck, the mutational variance as measured in MA experiments can be compared to evolution with natural selection in the wild (the intraspecific standing genetic variance and the interspecific divergence) to infer the role of natural selection. In this manner, we previously showed that P3.p division frequency likely evolved driven by its high mutational variance and under minimal selection (*Braendle et al., 2010*). Indeed, when either *C. elegans* or *C. briggsae* wild isolates are subjected to spontaneous mutation accumulation, P3.p cell fate had the highest phenotypic variance compared with the other five cells. P4.p showed the second highest mutational variance and standing genetic variance, yet an order of magnitude lower than P3.p (*Figure 1* and *Figure 1— figure supplement 1*; *Braendle et al., 2010*). Thus, in this system as for wing shape in drosophilids (*Houle and Fierst, 2013*; *Houle et al., 2017*) or mitotic spindle traits in *Caenorhabditids* (*Farhadifar et al., 2015*; *Farhadifar et al., 2016*), the mutational variance matches the evolutionary pattern, with the added advantage here of comparing homologous cells.

Here, we use MA lines to test whether P3.p fate evolvability is caused by a high mutation rate at few loci or by a broad mutational target affecting P3.p fate. To this end, we selected five MA lines showing P3.p fate divergence with the ancestral line. We combine whole-genome sequencing, genetic linkage analysis of the phenotype in recombinant lines and candidate testing through mutant and CRISPR genome editing to identify causal mutations and the corresponding loci. In each line, we found a single causal mutation. The five causal mutations are in five different genomic regions, are not associated to highly mutable sequences and are different in nature (two SNPs, one small deletion and two large deletions). Functionally, only one of them affected an expected gene involved in the Wnt pathway, a 'core' signaling pathway known to regulate Pn.p fusion to the epidermis in the L2 stage (*Pénigault and Félix, 2011b*). Two other loci encode general regulators of transcription and translation, while the two final loci lack functional annotation. We conclude that the fast evolutionary rate of change in P3.p cell fate may be explained by a broad mutational target for this trait.

## Results

### Choice of mutation accumulation (MA) lines

Estimating accurate frequencies for a binary trait requires a high number of individuals. We selected fifteen MA lines derived from two *C. briggsae* (HK104 and PB800) and two *C. elegans* (PB306 and N2) wild ancestors that had accumulated mutations for 250 generations (*Figure 2a* and *Figure 2— figure supplement 1*) with a putatively deviant P3.p division frequency from a previous study (*Braendle et al., 2010*). We phenotyped the selected lines again with their corresponding ancestral line with a large number of animals and in replicate experiments (see *Figure 2*, *Figure 2—figure supplement 1*, *Supplementary file 1* and Materials and methods). This led us to reduce the selection to six MA lines (two *C. briggsae* and four *C. elegans* lines) that displayed large differences in P3.p division frequency compared to their corresponding ancestral line, ranging from 19% to 53% (*Figure 2b*). These were MAL 211 and 296 derived from HK104 (*C. briggsae*), MAL 418, 450 and 488 derived from PB306 (*C. elegans*), and MAL516 from N2 (*C. elegans*).

### Whole-genome sequencing of ancestral and MA lines

We aimed to identify the spontaneous mutations that had appeared during the 250 generations of mutation accumulation with two main goals: (1) provide a reliable list of molecular markers for

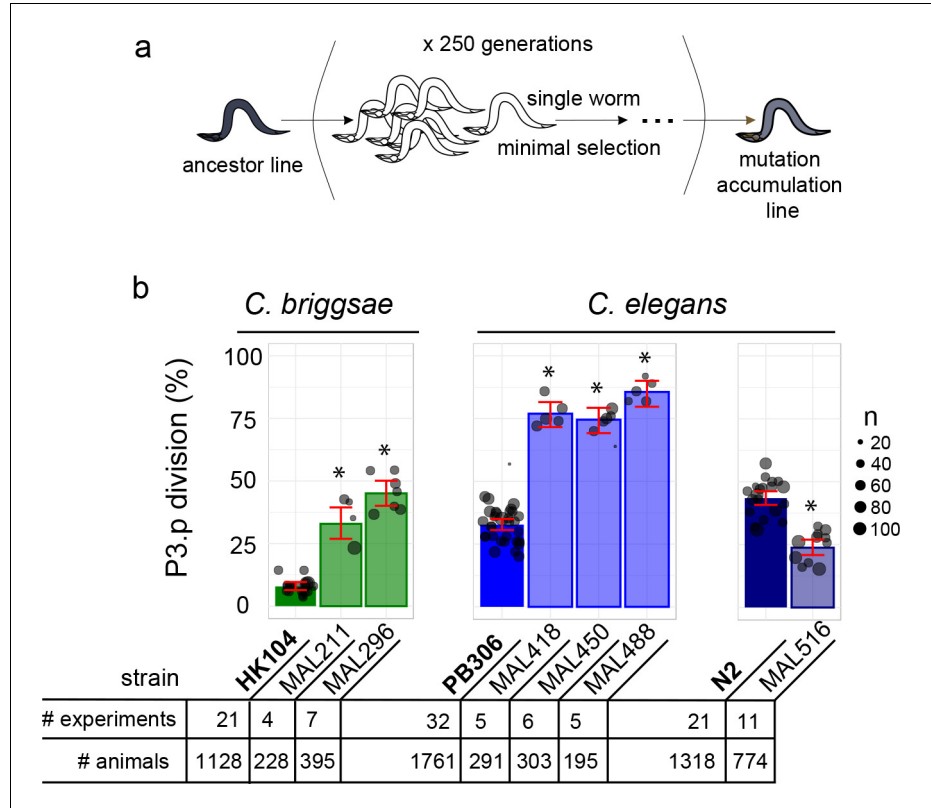

**Figure 2.** Choice of *Caenorhabditis* MA lines displaying evolution of P3.p cell fate compared to their ancestral line. (a) Schematic depiction of the generation of mutation accumulation (MA) lines. Starting from an ancestral line, each new generation is propagated through a single worm for many cycles (250 generations in the present case). This treatment with minimal selection at low population size increases the likelihood of fixing de novo spontaneous mutation by drift. (b) The panel of this study consists of three cohorts of ancestral lines and derived MA lines, one in the nematode species *C. briggsae* (derived from HK104 ancestor, colored in green in the figures) and two in *C. elegans* (derived from ancestors PB306 and N2, in blue). The bar charts represent the mean frequency of P3.p division for each strain in the three cohorts over several replicate experiments. Each dot represents an independent experiment, with dot size scaled to the number of scored individuals (n). The ancestral line is the leftmost strain (in bold). The number of independent experiments and individuals are indicated below the graphs. Stars indicate a significant difference with the ancestor line (Fisher's exact test) and error bars indicate 95% confidence intervals.

The online version of this article includes the following figure supplement(s) for figure 2:

**Figure supplement 1.** Selection of MA lines with evolution of P3.p cell fate compared to their ancestor line.

genetic linkage analysis; (2) find candidates for the causal mutation. Genomic DNA of the selected MA lines and their respective ancestor was sequenced at an average sequencing depth of 20x (*Supplementary file 2*). We used a combination of tools to cover a diversity of possible mutations (SNPs, short indels and structural variants). Prioritizing the first goal, we endeavoured to minimize false positive calls in two ways (see Materials and methods and *Figure 3—figure supplement 1*). First, we filtered out variants that were not unique to a MA line in a cohort derived from the same ancestor, so as to eliminate possible background variants that may have been missed in the ancestor. Such variants were particularly abundant in MA lines derived from the HK104 and PB306 ancestral backgrounds, which differ greatly from the reference genome of each species (AF16 for *C. briggsae* and N2 for *C. elegans,* respectively). Second, we excluded error-prone repeats from the short-variant analysis. These two filters excluded potential loci that could explain P3.p fate variation; in spite of this, the genetic linkage analysis should identify the chromosomal interval where the causal variant lies. A more sensitive variant analysis in this candidate interval would then be possible

if the causal variant was not found in the first stringent analysis (which turned out not to be required).

With this strategy, we listed 595 de novo mutations in the six MA lines, spread along the genome (*Figure 3—figure supplement 2* and *Supplementary file 3*). These mutations were mostly short (i.e shorter than the 100 bp read length) indels (341), SNPs (250), and four large deletions (*Figure 3— figure supplement 3*). We reliably used the SNPs from these calls directly as genetic markers: indeed, all but one over 60 SNP tested were validated by direct re-sequencing (*Figure 3—figure supplement 3*, see Materials and methods).

## Genetic mapping of the causal loci

Five of the six MA lines were further processed to genetically map the causal mutations affecting P3. p division frequency. The genetic mapping method relies on the same logic for all five MA lines (with some differences in the crossing schemes and selection strategies, see Materials and methods and *Figure 3—figure supplements 4–8*) generating several backcrossed lines, phenotyping and sorting them as 'ancestor-like' or 'MA-line-like' according to their phenotype (*Figure 3*; blue and red bars and dots, respectively) then genotyping them for a set of relevant de novo mutations identified above. Backcrossed lines were selfed for several generations to render them mostly homozygous. In all cases, the phenotype segregated as a single locus. A candidate genetic interval was defined as the minimum interval that bears the MA line genotype in all phenotypically MA-line-like backcrossed lines and the ancestral genotype in all phenotypically ancestor-like backcrossed lines. Serial back-crosses (once to four times) allowed to reduce the genetic interval, which still ranged from 4 to 15 Mbp (*Figure 3* and *Supplementary file 4*). Importantly, we identified intervals on four different chromosomes (I, III, IV and X) and two distinct regions on chromosome III. The genetic intervals were thus distinct in each line, excluding that recurrent mutations at a common locus could control the evolution of P3.p in the MA lines.

## Validation of the causal mutations by precise genome editing

The genetic intervals only contained few mutations (from 1 to 10). Predictions of functional impacts pointed to an obvious candidate lesion for each line. Four candidate lesions affected the coding region of a gene and the fifth was a large deletion spanning 10 genes (*Figure 3* and *Figure 4a*): two non-synonymous nucleotide substitutions in MAL 296 and 450, and deletions of 16, 1344 and 54,355 base pairs in MAL 488, 516 and 418, respectively.

The four single-gene mutations were validated by directly editing the genome of the ancestral line with CRISPR/Cas9-mediated homologous recombination technology to reproduce the mutation observed in the MA line (*Supplementary file 5*, see Materials and methods). In the case of the two non-synonymous nucleotide substitutions, we also introduced synonymous mutations in the guide RNA to avoid Cas9 re-cutting (*Supplementary file 5*) and hence used controls with the synonymous mutations but without the candidate non-synonymous substitution (*Figure 4b and c*). In the case of the 16 and 1344 base pairs (bp) deletions (*Figure 4d and e*), we provided a repair template that fully matched the sequence of the MA line in this region. In the case of the 54,355 bp deletion in MA line 418, we separately induced frameshifting indels via CRISPR/Cas9 in the coding region of seven genes within the interval and found that the deletion of one of them, *Y75B8A.8*, reproduced the P3. p phenotype of the MA line (*Figure 4f* and S11). This is in concordance with the analysis of different mutant lines for genes at this locus (*Figure 4—figure supplement 1c*). In all five cases, genome editing of the ancestor reproduced the change in P3.p division frequency observed in the MA line (*Figure 4*). These results were confirmed by phenotyping two independent CRISPR lines (*Figure 4*) and independent alleles of the same gene (*Figure 4—figure supplement 1c*).

The induced mutations also reproduced pleiotropic alterations of vulva traits or other phenotypes that were co-segregating with P3.p behavior during the backcrosses (*Supplementary file 6*) – while some other phenotypes were eliminated by backcrossing. These results demonstrated that the five candidate mutations identified by genetic linkage analysis were necessary and sufficient to explain the evolution of P3.p division frequency.

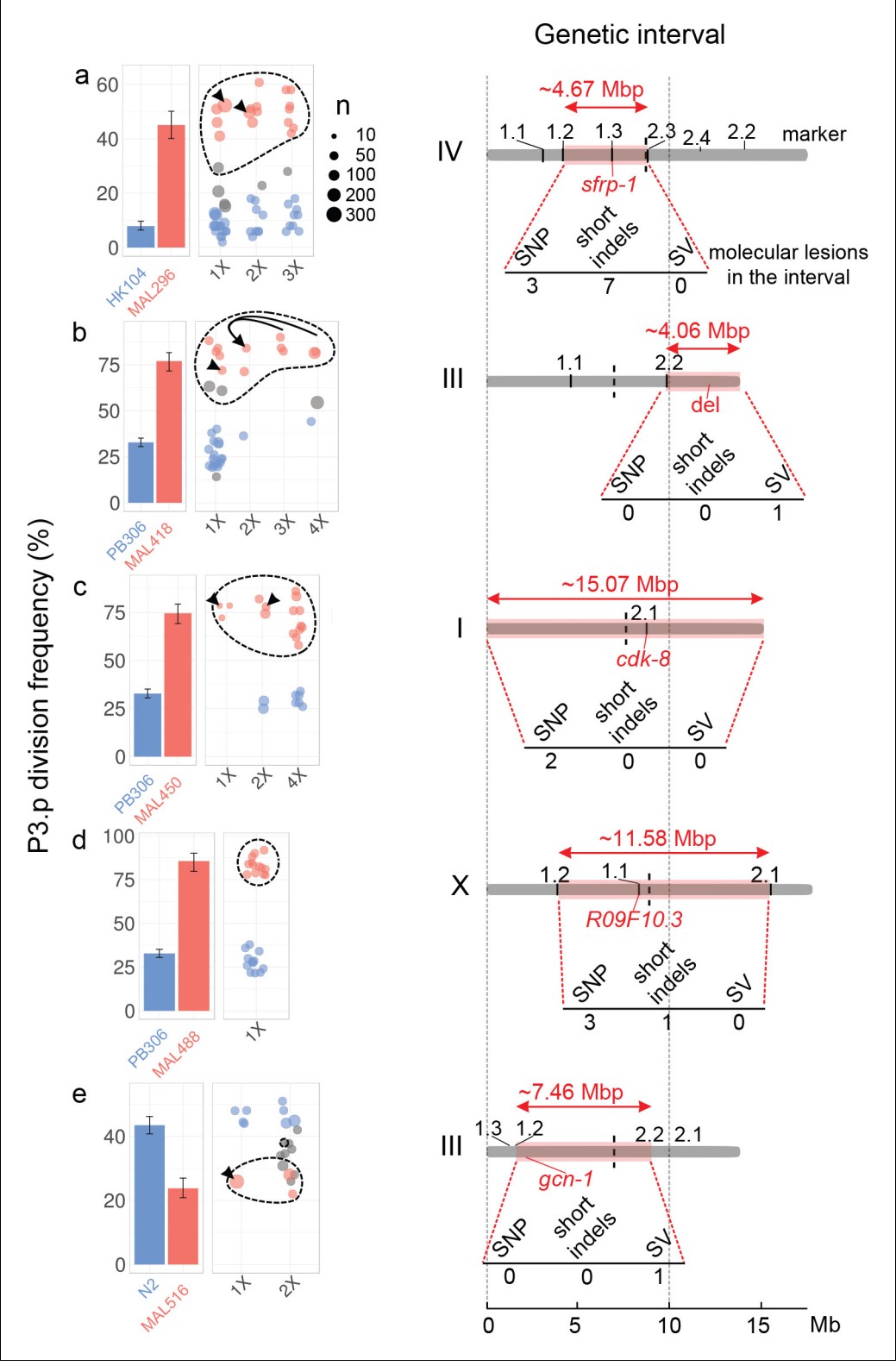

**Figure 3.** The evolution of P3.p fate maps to a single locus in each mutation accumulation line, each in a different genomic region. For each panel (a–e), plots on the left indicate the frequency of P3.p division for the ancestral line, the MA line and successive rounds of backcrosses (designated 1x to 4x, see Materials and methods). Data for ancestor and MA lines in the leftmost panel are those shown in *Figure 2b*. Error bars are 95% confidence intervals. Each dot is a different backcrossed line, the size of which indicates the number of animals assayed (n, several

*Figure 3 continued*

independent replicates may be pooled). Dot colors correspond to statistical groups determined by post-hoc analysis of pair-wise Fisher's tests among backcrossed lines (fdr level: 0.05: red dots are not different from the parent MA line but different from ancestor, blue dots are not different from ancestor but different from MA lines and gray dots are either different from both or not different from either. Dashed lines indicate the backcrossed lines that carry the candidate mutation in the mapping interval. Black arrowheads point to the strain that was used as a parent for the next backcross. In panel b, the same 2x parent was used to independently yield 3x and 4x backcross lines, the latter through crossing the hybrid males to the parental line. Diagrams on the right indicate the position and size of the genetic interval (red rectangle) on the chromosome (gray bar), as identified by combining P3.p scores and genotyping data. The identifiers indicated above the chromosome ('1.1', '1.2', etc.) correspond to the pyrosequencing markers. The number of de novo mutations predicted in each interval is indicated below each diagram. 'SV': structural variant. The position of the causal gene (or mutation) is indicated in red.

The online version of this article includes the following figure supplement(s) for figure 3:

**Figure supplement 1.** Schematic workflow used for variant discovery in the sequenced genomes.
**Figure supplement 2.** Genome-wide distribution of spontaneous mutations accumulated in the MA lines sequenced for this study.
**Figure supplement 3.** Variant discovery and validation in mutation accumulation lines.
**Figure supplement 4.** Crossing scheme and selection strategies used to backcross MA line 296 into its ancestral line HK104.
**Figure supplement 5.** Crossing scheme and selection strategies used to backcross MA line 418 into its ancestral line PB306.
**Figure supplement 6.** Crossing scheme and selection strategies used to backcross MA line 450 into its ancestral line PB306.
**Figure supplement 7.** Crossing scheme and selection strategies used to backcross MA line 488 into its ancestral line PB306.
**Figure supplement 8.** Crossing scheme and selection strategies used to backcross MA line 516 into its ancestral line N2.

## Molecular nature of the causal mutations and mutation rates at these loci

The molecular nature of the five mutations was diverse (*Figures 4a* and *5a*): two non-synonymous single-nucleotide substitutions, a small 16 bp deletion and two larger deletions of 1,344 bp and 54,355 bp. The substitutions are a T-to-G transversion and a T-to-C transition, which are not the most frequent substitution types in *Caenorhabditis* spontaneous mutation accumulation lines (*Denver et al., 2012*). Considering the three-bp motif (with the mutant base at the 3' end) (*Saxena et al., 2019*), the corresponding motifs (AT**T** and AG**T**, respectively) were not reported to be those with the highest spontaneous mutation rates either. Small deletions have lower mutation rates than single-nucleotide substitutions (*Saxena et al., 2019*; *Konrad et al., 2019*). As for the large deletions, they appear less frequent that large insertions/gene duplications (*Konrad et al., 2018*). Thus, these five mutations do not point to particularly frequent types of mutation.

Next, we analysed the surrounding sequences of the causal mutations and their local and global genomic contexts and found no common element among the five mutations: they lie in regions with different GC contents (from 16% to 50% in a 50 bp window centered on the causal mutation), in regions either rich or poor in repeats, in chromosome arms or centres (*Figure 5b* and *Figure 5—figure supplement 1a–i*). Repeats are associated to higher mutation rates (*Heale and Petes, 1995*; *McDonald et al., 2011*). In sequence data of other *C. elegans* spontaneous MA lines (*Saxena et al., 2019*), we indeed found an overrepresentation of mutations in repeated sequences: 42% of mutations (n = 3469) were found in repeated sequences that represent 20% of the genome ($X^2$-test: p-value<$2.2\times10^{-16}$; however, note that false-calling rates are expected to be higher in repeats). Of the causal mutations, the two substitutions and the 16 bp deletion do not lie in repeats. The 3' breakpoint of the large 54,355 bp deletion lies within a repeat (*Figure 5b*), but is far away from the causative gene *Y75B8A.8* that lies at the 5' end of this 54 kb deletion (*Figure 4—figure supplement 1*). The other large deletion, however, lies in an AT-rich region (two introns of the *gcn-1* gene) that may be classified as 'tandem and inverted repeats' and the two breakpoints correspond to a 20 bp direct repeat with two mismatches (*Figure 5—figure supplement 1j*).

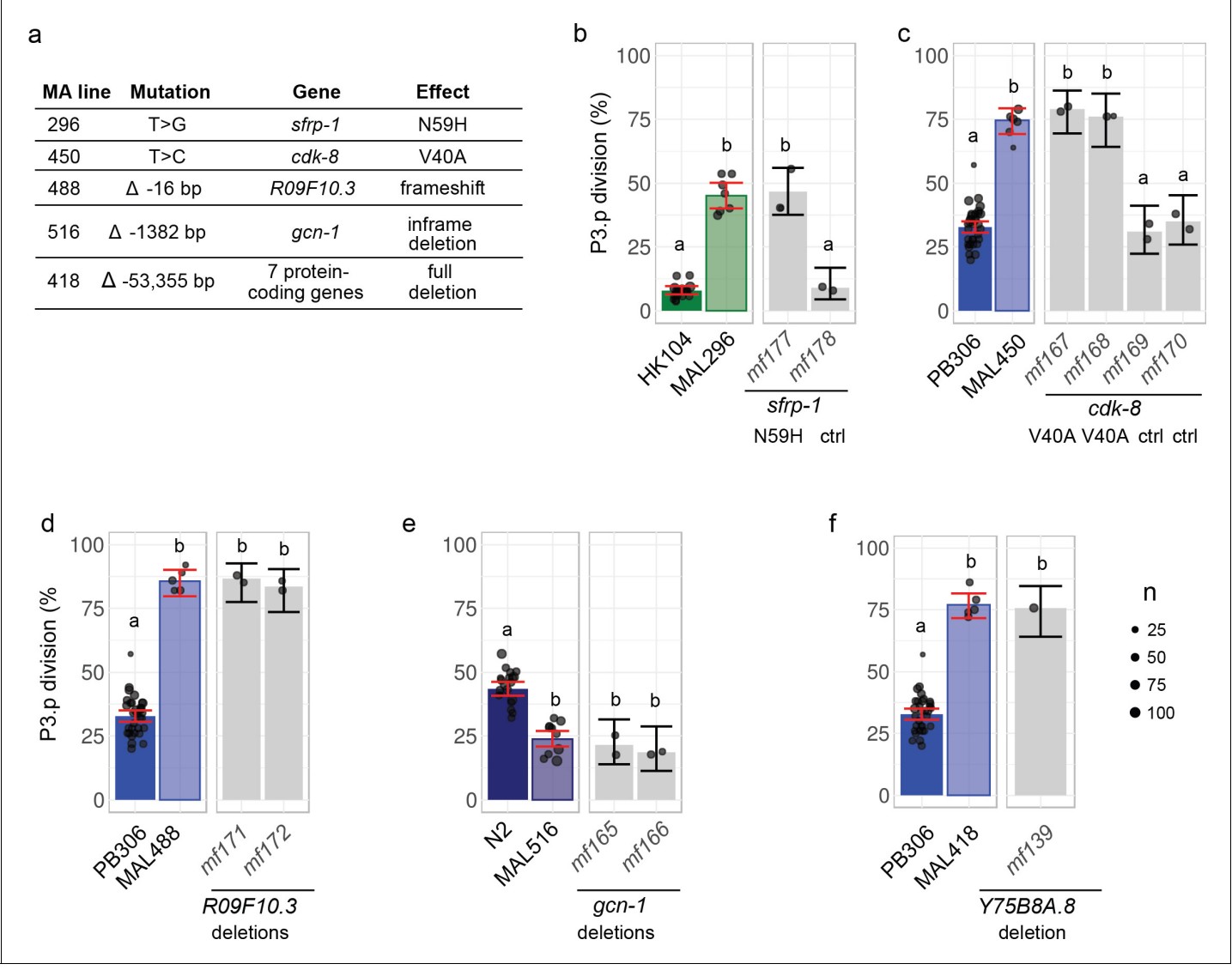

**Figure 4.** Validation by precise genome editing of candidate causal mutations responsible for P3.p cell fate evolution in MA lines. (a) Summary table of the molecular nature, underlying gene and molecular effect of the candidate mutations. (b) P3.p division frequency after editing the *sfrp-1* locus in ancestor HK104 with a repair template coding only for synonymous substitutions (*mf178*) or introducing the N59H substitution as well (*mf177*). (c) P3.p division frequency after editing the *cdk-8* locus in ancestor PB306 with a repair template coding for synonymous substitutions only (independent edits *mf169* and *mf170*) or introducing the V40A substitution as well (independent edits *mf167* and *mf168*). (d) P3.p division frequency after editing the *R09F10.3* locus in ancestor PB306 to reproduce the exact same 16 bp deletion as in MA line 488 (independent edits *mf171* and *mf172*). (e) P3.p division frequency after editing the *gcn-1* locus in ancestor N2 to reproduce the exact same 1344 bp deletion as in MA line 516 (independent edits *mf165* and *mf166*). (f) P3.p division frequency after deleting the entire *Y75B8A.8* locus in ancestor PB306. Each dot is an independent experiment, with dot size scaled to the number of scored individuals(n). The bar is the mean frequency obtained by pooling all replicates; error bars indicate 95% confidence intervals. For each graph, leftmost panels provide the scores of ancestor and MA lines as reference (identical data to *Figure 2b*). Different letters indicate a significant difference (Fisher's exact test, fdr level: 0.05).

The online version of this article includes the following figure supplement(s) for figure 4:

**Figure supplement 1.** P3.p division frequency in mutants of individual genes within the large deletion of MA line 418.

**Figure supplement 2.** P3.p cell fate in different mutants related to the candidate mutation found in MA lines 296 (a), 450 (c) and 516 (e).

We therefore directly inquired whether this deletion (and the other mutations) occurred recurrently at a detectable frequency by analyzing sequence data of other MA lines (*Saxena et al., 2019*: 75 other MA lines, 3469 nuclear mutations). We did not find any other mutation at the

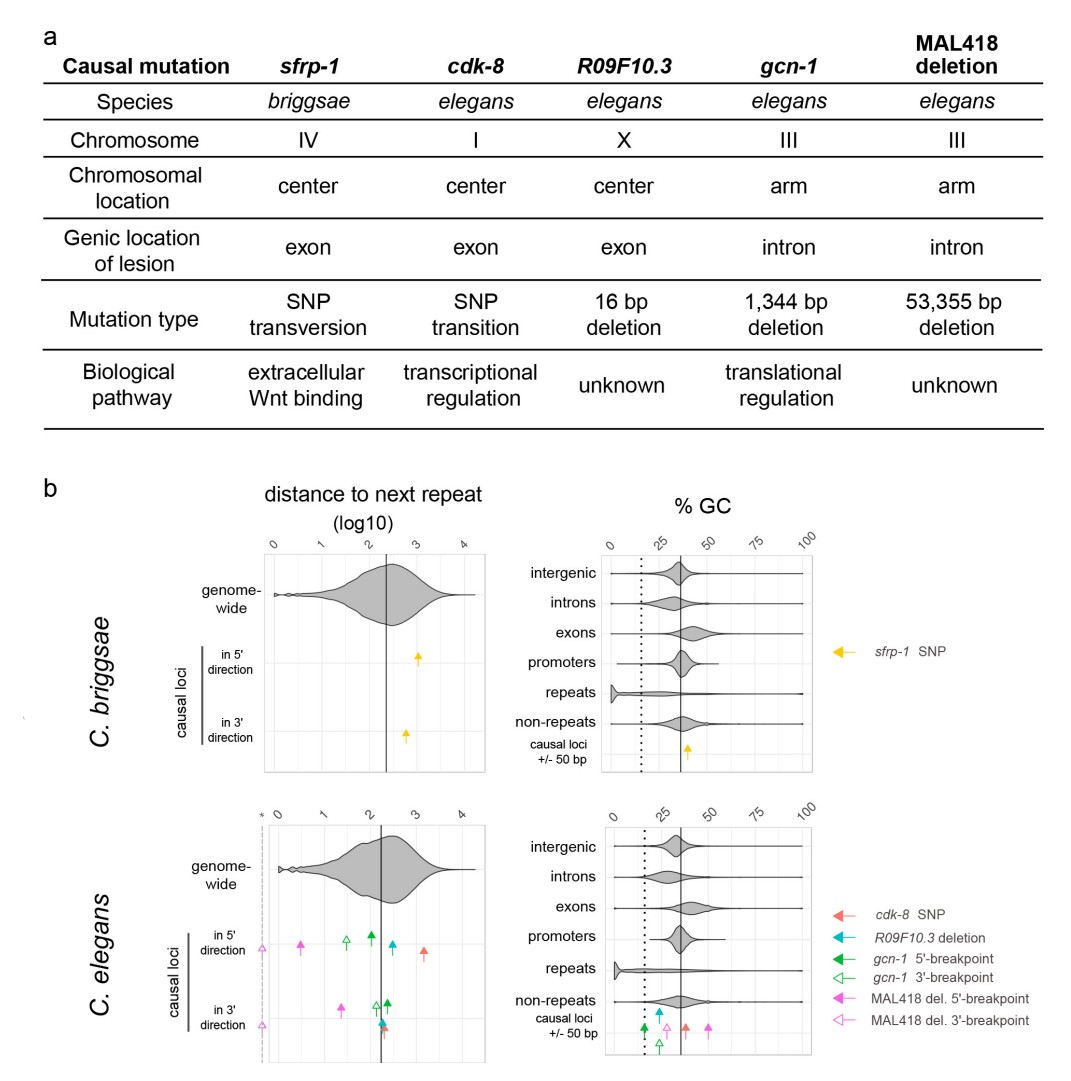

**Figure 5.** The causal mutations and underlying genes are diverse and do not correspond to repeats. (a) The five causal mutations correspond to a diversity of chromosomal locations, molecular lesions, genes and biochemical pathways. (b) The five causal mutations correspond to a diversity of locations relative to repeats and GC content. Upper and lower panels show data from *C. briggsae* and *C. elegans*, respectively. For each graph, violin plots show the distribution for genomic sequences, while colored arrows indicate the value for each causal locus. In the left panels, arrows indicate the distance in base pairs (log10) of each causal locus to the closest repeat in 5' or in 3', while the violin plot shows the distribution of all inter-repeat distances in the genome. The vertical line corresponds to the genome median value. For large deletions, 5' and 3' breakpoints have been considered as two distinct loci. The dashed gray line marked with a star in the x-axis indicates zero values for the deletion 3' end lying within a repeat. Note that the *Y75B8A.8* gene lies towards the 5' end of the large deletion in MA line 418, thus the repeat corresponding to the 3' end is far from the gene. In the right panels, the percentage of GC in a small 50 bp window centered around each causal locus is compared to the GC values of different types of genomic sequences. The plain vertical line is the GC content of the entire genome and the dashed vertical line is the median GC content of repeats. The online version of this article includes the following figure supplement(s) for figure 5:

**Figure supplement 1.** Global and local sequence context of causal mutations.

corresponding positions and the closest mutations were at least 4 kb away (*Supplementary file 7*). This result excludes an extremely high mutation rate at the position of the five causal mutations.

However, the size of the MA line dataset limits our ability to detect quantitative differences in mutation rates that could be significant at evolutionary time scales. We thus used two further data-sets with abundant variation: the Million Mutation Project (MMP, *Thompson et al., 2013*) and the *Caenorhabditis elegans* Natural Diversity Resource (CeNDR, *Cook et al., 2017*). The MMP dataset provides enough power, but is derived from lines after chemical and/or ultraviolet mutagenesis

aiming at producing deletions (2007 strains with about 400 mutations each, *Thompson et al., 2013*). None of the five nucleotide positions (breakpoints for deletions) were mutated in this dataset (*Supplementary file 7*). One deletion was found in *gcn-1* but breakpoints do not match the identified direct repeats. The caveat of using the MMP dataset is that the pattern of artificially induced mutations may differ from that of spontaneous mutation. Second, we explored the *C. elegans* natural diversity (almost 3 million genomic variations from 766 wild strains; *Cook et al., 2017*), and none of the positions (the breakpoints for deletions) were mutated either (*Supplementary file 7*). The caveat of using this dataset is that selection has acted on the polymorphism pattern; note however that the *gcn-1* repeats lie in intronic regions where mutations may have less functional impact (*Figure 5—figure supplement 1j*). We thus conclude that the five identified mutations are not in mutational hotspots.

We next wondered whether the underlying genes (rather than the precise positions) - the first level of sequence to phenotype mapping - could display higher mutation rates. The mutation rate of a gene depends on its length and the mutation rate of its sub-sequences. Among the five genes, *gcn-1* and to a lesser extent *Y75B8A.8* stand out as large genes (measured from 5'UTR to 3'UTR, including introns): they are the 10th and 841st longest genes among the 21,803 *C. elegans* protein-coding genes, respectively (*Figure 6—figure supplement 1a*). Their total repeat content is longer, mainly in introns for *gcn-1* and in both introns and exons for Y75B8A.8 (*Figure 6—figure supplement 1b*).

In the 75 *C. elegans* MA lines we analyzed, none of the five genes showed a second hit in their exons, even though some other genes were recurrently mutated, including in exons (*Figure 6a*). In the MMP and CeNDR, genes accumulate mutations as predicted by their length (*Figure 6b,c*), thus *gcn-1* is often hit. *gcn-1* retains natural variations at a higher rate than the average of genes, due to introns, where variations are less likely to impact protein function (*Figure 6c*). From these data, we concluded that the five causative genes do not present particularly high mutation rates given their length.

If only polymorphisms annotated with a predicted high or moderate impact on protein function are taken into account, most genomes of wild isolates at CeNDR do not bear such variants for *sfrp-1* and *cdk-8* (99% and 97% respectively, n = 330), likely due to purifying selection (*Figure 6—figure supplement 1c*). Non-synonymous polymorphisms are more frequent for the three other causative genes (*Figure 6—figure supplement 1c*). This suggests that variations in the protein sequence corresponding to these three genes do not generate strongly counter-selected phenotypes in nature. Further experiments are required to quantify how much this natural polymorphism contributes to the high standing genetic variance measured for P3.p (*Figure 1—figure supplement 1*).

## Relations between the causative genes and the effects on P3.p phenotype

We then aimed to understand how these different loci affect P3.p cell fate by analysing the nature of the underlying genes. One of the five genes, *sfrp-1*, was an obvious candidate regulating the Wnt pathway; the other four were not.

SFRP-1 (Secreted Frizzled Receptor Protein-1, mutated in *C. briggsae* MA line 296) is a highly conserved secreted Frizzled protein that inhibits Wnt signaling by sequestering Wnts. In *C. elegans*, the *sfrp-1* gene is expressed in the anterior part of the nematode and the protein counter-acts the effect of posteriorly secreted Wnts (*Harterink et al., 2011*; *Figure 4—figure supplement 2b*). Since P3.p is highly sensitive to the posterior Wnt gradient (*Pénigault and Félix, 2011b*), loss of *sfrp-1* should increase the frequency of P3.p division. Indeed, we observed an increase in P3.p division frequency for *C. briggsae* MA line 296 and the corresponding *sfrp-1* genome edits compared to the HK104 ancestor (*Figure 4b*). Using an available null mutant line in *C. elegans*, we showed that the effect of *sfrp-1* on P3.p division is conserved in both species, and opposite to the effect of a decrease in canonical Wnt signaling through a null *bar-1* mutation (*Figure 4—figure supplement 2a*). The mutation in MA line 296 is a missense in the cystein-rich Frizzled domain that binds the Wnt ligand, changing a conserved asparagine into a histidine (*Figure 4a*).

The *cdk-8* gene (cyclin-dependent kinase-8, mutated in *C. elegans* MA line 450) codes for a subunit of the Mediator complex. This conserved eukaryotic multiprotein complex interacts with chromatin, transcription factors and the RNA Polymerase II machinery and regulates the transcription of many genes (*Grants et al., 2015*; *Angeles-Albores and Sternberg, 2018*). Its specificity of action

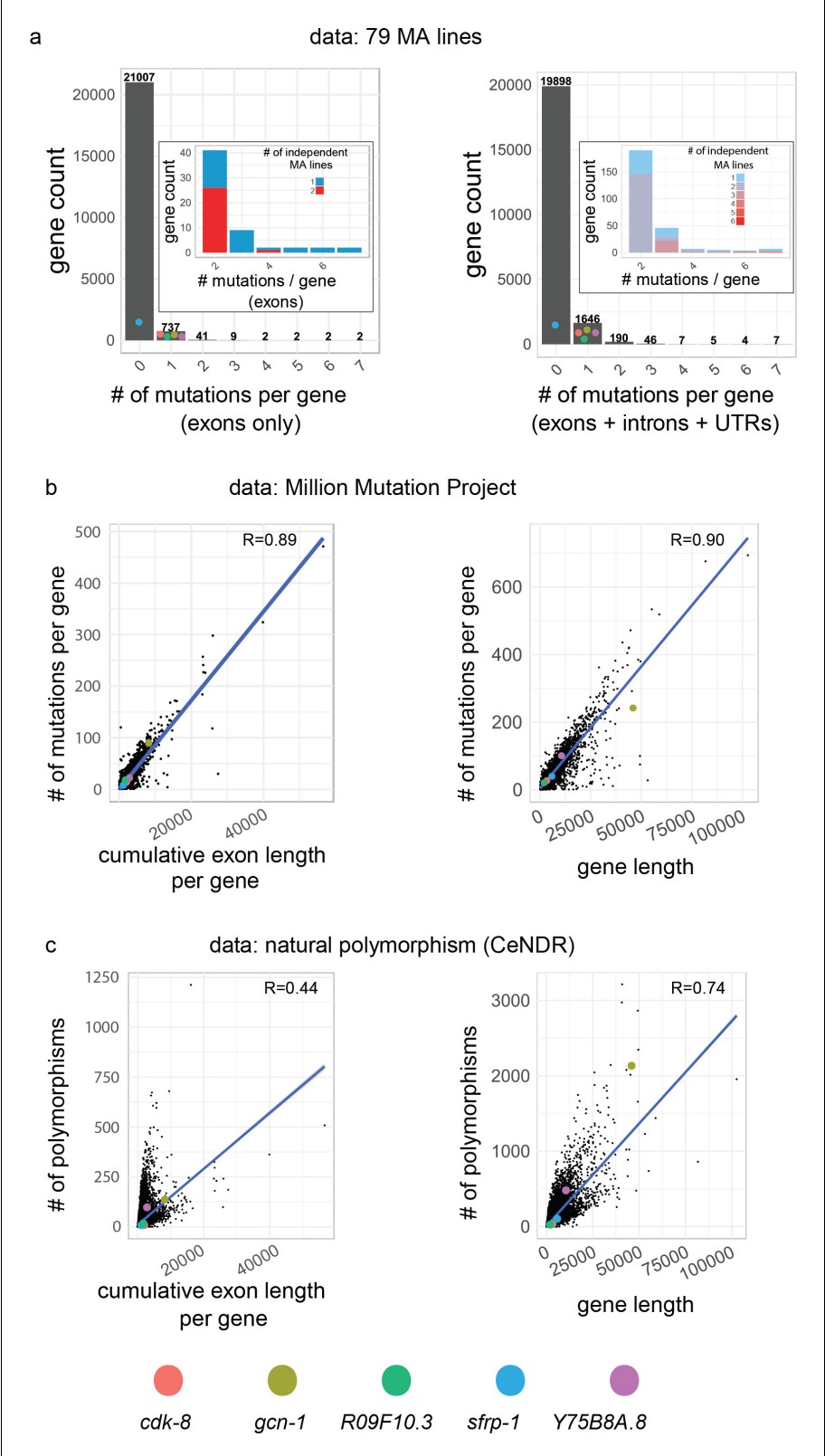

**Figure 6.** Mutational properties of the five causative genes. (a) Distribution of number of hits in protein-coding genes in MA lines (this study + 75 lines from *Saxena et al., 2019*). Throughout the figure, the left panels show cumulative length and mutations of exons only, while the right panels show the length and mutations of genes, defined as the primary transcript sequence (including exons, introns and untranslated regions). Inset focuses on genes with at least two hits, the color code indicating whether hits were found in the same or independent MA lines. Colored dots indicate the value for each

*Figure 6 continued*

causative gene of this study, which were hit only once, except *sfrp-1* which was not hit in the *C. elegans* data set (it was found in a *C. briggsae* MA line). (**b**) Correlation between the cumulative exon length (left) and gene length (right) and the number of corresponding mutations in the Million Mutation Project (*Thompson et al., 2013*). (**e**) Correlation between the cumulative exon length (left) and gene length (right) and the corresponding number of polymorphic sites, from data from the *Caenorhabditis* Natural Diversity Resource (CeNDR; *Cook et al., 2017*). In (**b,c**), R is the Pearson's correlation coefficient (p-value$<2.10^{-16}$ in all cases).

The online version of this article includes the following figure supplement(s) for figure 6:

**Figure supplement 1.** Mutational properties of causative genes.

on transcription is controlled by distinct dissociable subunits, such as the CDK-8 module. In *C. elegans*, the CDK-8 module acts in a highly pleiotropic fashion yet a P3.p division frequency phenotype was not previously reported. In the ventral epidermis, the CDK-8 module was shown to act at many other steps, contributing in the L1 stage to the fusion to hyp7 of anterior and posterior Pn.p cells (such as P2.p and P9.p) (*Yoda et al., 2005*), to the block of division of all VPCs in the L2 stage (*Clayton et al., 2008*) and to the level of induction of 2° and 1° VPC fates via cell-autonomous repression of EGF and Notch signalling in the L3 stage; these activities being mostly revealed in a sensitized genetic background (*Moghal et al., 2003*; *Grants et al., 2016*; *Underwood et al., 2017*). We found that mutation in three other genes encoding components of the CDK-8 module also increased P3.p division frequency in an otherwise wild-type genetic background (*Figure 4—figure supplement 2c,d*): *cic-1*, *dpy-22/mdt-12* and *let-19/mdt-13*. The valine-to-alanine substitution in the protein kinase domain found in MA line 450 likely causes a strong reduction-of-function, since the phenotypes such as dumpy animals or P3.p division frequency were indistinguishable from those in animals bearing the null deletion allele *cdk-8(tm1238)* (*Grants et al., 2016*; *Figure 4—figure supplement 2c* and *Supplementary file 7*). To test whether CDK-8 acts independently of the Wnt signalling pathway to modulate P3.p division frequency, we performed epistatic analysis by combining null mutants of *cdk-8* and *bar-1*. The double mutants showed an intermediate level of P3.p division frequency (*Figure 4—figure supplement 2c*), thus *cdk-8* was not epistatic to *bar-1* suggesting that CDK-8 functions independently of the Wnt signalling pathway. In sum, CDK-8 is part of a large complex that is a general regulator of transcription; its mutation, although not lethal, is likely to affect many processes that are sensitive to the level of transcription of one or several of the many downstream genes.

The *gcn-1* gene (homolog of yeast General Control Non-derepressible) is a large protein of 2651 amino-acids (aa) including several Armadillo repeats, conserved throughout eukaryotes. The GCN-1 protein is involved in translational control. GCN-1 promotes the phosphorylation of the eukaryotic initiation factor eIF2α (*Nukazuka et al., 2008*), which is thought to globally repress translation while activating expression of a few specific genes in many eukaryotes. This pathway is known to be active under various environmental stresses and to regulate global metabolic homeostasis (*Rousakis et al., 2013*; *Figure 4—figure supplement 2f*). Local repression of this pathway by semaphorin signalling is required for *C. elegans* male ray morphogenesis (*Nukazuka et al., 2008*). The *gcn-1* mutation in the MA line 516 deletes the entire 21st exon and flanking intronic regions removing a part of the translation elongation factor three protein domain that is required for the efficient phosphorylation of eukaryotic initiation factor 2 (*Hirose and Horvitz, 2014*). From comparison with another partial deletion allele, *gcn-1(nc40)*, the MA line mutation is likely a reduction-of-function allele (*Figure 4—figure supplement 2e*). GCN-1 had not been involved so far in the regulation of P3.p division.

Little is known about the two last genes. *Y75B8A.8*, entirely deleted in MA line 418, codes for a 715-aa protein lacking any known functional domain and homology outside nematodes. The protein bears features of intrinsically disordered proteins, including polyglutamine stretches in the N-terminal half (https://wormbase.org/species/c_elegans/protein/CE34135#065-−10). The homologous protein in the parasitic nematode *Haemonchus contortus* is found in excretory and secretory products and is able to bind the interleukin IL2 of its mammalian host (*Wang et al., 2019*). In *C. elegans*, the 3'UTR of *Y75B8A.8* regulates RNA editing of the ADSR-2 mRNA (*Wheeler et al., 2015*; *Washburn and Hundley, 2016*). This gene was not known to affect Pn.p cell development.

Finally, *R09F10.3* is a 468-aa protein with a weak similarity to the Mediator subunit MED27 at its C-terminus and no detectable similarity of the N-terminal part (https://wormbase.org/species/c_

elegans/protein/CE33810#065-−10). The short deletion in MA line 488 induces a frameshift and an early stop codon truncating more than 40% of the protein length. This gene was not known to affect Pn.p cell development.

## Discussion

In this study, we report the first identification of mutations underlying a trait's high mutational variance in mutation accumulation lines. Using the highly tractable development of *Caenorhabditis* nematodes at the cellular scale, their powerful genetics and the recent advances in genome editing, we could precisely characterize mutational events that drove the fast evolution of a trait in a controlled evolutionary experiment. Our random sampling of mutations driving the evolution in P3.p division frequency in MA lines hit five different genes with no signature of high mutation rates, which could be connected to at least three different functional modules: Wnt signalling, transcriptional control by the Mediator complex and translational control through GCN-1. A the level of the genes, one of them (*gcn-1*) is particularly long so it is likely to be the target of mutations, whereas three of them are quite short.

Using this quantitative genetics approach, we were able to find new regulators of P3.p developmental fate that are available for further developmental studies. This is a small sample of possible mutations and already demonstrates that the cellular process of P3.p division is sensitive to variation in a larger number of genes and pathways. We conclude that the higher mutational variance of P3.p cell division is not specifically due to the higher mutability of particular DNA sequences and cannot be predicted from the genome sequence. Instead, it is a consequence of a broad mutational target impacting this cell fate specification, thus to the developmental context controlling the decision of P3.p to either fuse with hyp7 in the L2 stage or to further divide in the L3 stage. This result on the role of genotype-phenotype mapping in the evolutionary rate has broad implications in evolutionary biology of any organism (unicellular, multicellular, viruses). In addition, mutational effects on the phenotype are of obvious consequences in genetic disease and in the phenotypic progression of cancerous tumors.

An obvious further question is whether the mutations found in MA lines are representative of those responsible for P3.p evolution in natural populations. At least three out of five identified mutations affected important fitness-related traits such as body morphology or fertility, as well as other vulva traits (albeit at much lower frequency than changes observed for P3.p, *Supplementary file 6*). The fast evolutionary rate of change in P3.p cell fate could then be driven: (1) by the subset of mutations with little pleiotropy in the corresponding genetic background (different from that tested here) or with pleiotropic effects that can be soon compensated for, or (2) by pleiotropic mutations that can be selected positively for their effect in other tissues (*Duveau and Félix, 2012*). Among 'target' genes, the most polymorphic in natural populations could be a reservoir of natural mutations affecting P3.p (*Figure 6—figure supplement 1c*). We also note that we selected large-effect mutations on purpose to ease the genetic mapping. It is possible that small-effect mutations would appear less pleiotropic. In any case, the diversity of functional pathways identified in this study offers opportunities to generate such non-pleiotropic small-effect mutations. A prediction from our present work is that mapping genetic determinants of P3.p division frequency in natural isolates should identify many different small-effect loci, possibly involving more functional pathways. Such an experiment remains however practically difficult to carry out, given the binary nature of the trait that imposes to score the phenotype of numerous isogenic animals to estimate reliable frequencies, the current low-throughput phenotyping and the highly multigenic nature of the trait.

From a developmental perspective, the reason why P3.p cell fate has such a broad mutational target likely lies in the sensitivity of this cell fate decision to small quantitative alterations in many biochemical pathway activities or in this cell's position. Indeed, we previously showed that P3.p division frequency is sensitive to halving the dose of either of the two Wnt ligands that are secreted from the posterior end of the animal (*Pénigault and Félix, 2011b*). P3.p is located at the fading end of the posterior-to-anterior Wnt gradient and may therefore often receive a Wnt dose that is below the threshold required for its division, while P4.p and the most posterior cells are more robustly induced. In addition to core Wnt pathway genes, mutations acting on other biochemical pathways and in other cells (e.g. neurons; see *Modzelewska et al., 2013*) could affect P3.p fate if they resulted in small variations in Wnt gradient levels, cellular position within the gradient, or interpretation of the

gradient. In addition, the sustained expression of the Hox gene *lin-39* is required to prevent Pn.p cell fusion in the L2 stage (*Eisenmann and Kim, 2000*), independently of Wnt signalling (*Pénigault and Félix, 2011b*). Hox gene regulation may be a further mutational target underlying the high mutational variance of P3.p fate. In summary, P3.p is located at a very sensitive position that results in its developmental fate being highly sensitive to stochastic, environmental and genetic variation (*Braendle and Félix, 2008*). The broad mutational target that we find here is consistent with this developmental sensitivity.

Variability of cell fates among the six vulva precursors evolved significantly among rhabditids. In another genus of the same family, *Oscheius*, P3.p cell fate is not highly variable (it does not divide), whereas P4.p and P8.p cell fates vary extensively both within and among species (*Delattre and Félix, 2001*). It would be interesting to test whether these different evolutionary rates correspond to an evolution in the respective mutational variances explained by broader mutational targets. The assembly and annotation of the *Oscheius tipulae* genome makes now possible to identify functional pathways involved in development of this species (*Besnard et al., 2017*; *Vargas-Velazquez et al., 2019*). This would offer a way to study how the evolution of developmental mechanisms correlates with the evolution of mutational variance and ultimately results in the evolution of evolutionary rates.

# Materials and methods

**Key resources table**

| Reagent type (species) or resource | Designation | Source or reference | Identifiers | Additional information |
|---|---|---|---|---|
| Gene (*C. elegans*) | *cdk-8* | WormBase | WBGene00000409 | |
| Gene (*C. elegans*) | *gcn-1* | WormBase | WBGene00021697 | |
| Gene (*C. elegans*) | *R09F10.3* | WormBase | WBGene00019987 | |
| Gene (*C. elegans*) | *Y75B8A.8* | WormBase | WBGene00013545 | |
| Gene (*C. elegans*) | *sfrp-1* | WormBase | WBGene00022242 | |
| Gene (*C. briggsae*) | *Cbr-sfrp-1* | WormBase | WBGene00027904 | |
| Strain, strain background (*C. briggsae*) | HK104 | DOI:10.1073/pnas.0406056102 | HK104[CB] WormBase ID: WBStrain00041077 | Wild isolate. Ancestor strain of MA lines. |
| Strain, strain background (*C. briggsae*) | MAL211 | DOI:10.1371/journal.pgen.1000877 | MAL211 | Mutation Accumulation line (250 generations) |
| Strain, strain background (*C. briggsae*) | MAL296 | DOI:10.1371/journal.pgen.1000877 | MAL296 | Mutation Accumulation line (250 generations) |
| Strain, strain background (*C. elegans*) | N2 | DOI:10.1073/pnas.0406056102 | N2[CB] WormBase ID: WBStrain00000001 | Lab reference strain. Ancestor strain of MA lines. |
| Strain, strain background (*C. elegans*) | MAL516 | DOI:10.1371/journal.pgen.1000877 | MAL516 | Mutation Accumulation line (250 generations) |
| Strain, strain background (*C. elegans*) | PB306 | DOI:10.1073/pnas.0406056102 | PB306[CB] WormBase ID: WBStrain00030546 | Lab reference strain. Ancestor strain of MA lines. |
| Strain, strain background (*C. elegans*) | MAL418 | DOI:10.1371/journal.pgen.1000877 | MAL418 | Mutation Accumulation line (250 generations) |
| Strain, strain background (*C. elegans*) | MAL450 | DOI:10.1371/journal.pgen.1000877 | MAL450 | Mutation Accumulation line (250 generations) |
| Strain, strain background (*C. elegans*) | MAL488 | DOI:10.1371/journal.pgen.1000877 | MAL488 | Mutation Accumulation line (250 generations) |

*Continued on next page*

*Continued*

| Reagent type (species) or resource | Designation | Source or reference | Identifiers | Additional information |
|---|---|---|---|---|
| Genetic reagent (*C. briggsae*) | *Cbr-sfrp-1* (*mf177*) | this paper | JU3707 | N59H edited allele. Background strain: HK104[CB] cf Suppl. File 5. |
| Genetic reagent (*C. briggsae*) | *Cbr-sfrp-1* (*mf178*) | this paper | JU3708 | Control edited allele with synonymous mutations. Background strain: HK104[CB] cf Suppl. File 5. |
| Genetic reagent (*C. elegans*) | *gcn-1(mf165)* | this paper | JU3641 | Precise deletion of exon 21 as in MAL516. Background strain: N2[CB] cf Suppl. File 5. |
| Genetic reagent (*C. elegans*) | *gcn-1(mf166)* | this paper | JU3642 | Precise deletion of exon 21 as in MAL516. Background strain: N2[CB] cf Suppl. File 5. |
| Genetic reagent (*C. elegans*) | *cdk-8(mf167)* | this paper | JU3643 | V40A edited allele. Background strain: PB306[CB] cf Suppl. File 5. |
| Genetic reagent (*C. elegans*) | *cdk-8(mf168)* | this paper | JU3644 | V40A edited allele. Background strain: PB306[CB] cf Suppl. File 5. |
| Genetic reagent (*C. elegans*) | *cdk-8(mf169)* | this paper | JU3645 | Control edited allele with synonymous mutations. Background strain: PB306[CB] cf Suppl. File 5. |
| Genetic reagent (*C. elegans*) | *cdk-8(mf170)* | this paper | JU3646 | Control edited allele with synonymous mutations. Background strain: PB306[CB] cf Suppl. File 5. |
| Genetic reagent (*C. elegans*) | *R09F10.3(mf171)* | this paper | JU3647 | 16 bp deletion as in MAL488. Background strain: PB306[CB] cf Suppl. File 5. |
| Genetic reagent (*C. elegans*) | *R09F10.3(mf172)* | this paper | JU3648 | 16 bp deletion as in MAL488. Background strain: PB306[CB] cf Suppl. File 5. |
| Genetic reagent (*C. elegans*) | *Y75B8A.8(mf139)* | this paper | JU3357 | Deletion in exon 3. Background strain: PB306[CB] cf Suppl. File 5. |
| Recombinant DNA reagent | pJA58 (plasmid) | Addgene | Addgene: Plasmid #59933 | |
| Sequence-based reagent | Alt-R CRISPR-Cas9 tracrRNA | IDT | Cat#: 1072533 | |
| Sequence-based reagent | crRNA (for *cdk-8*; *gcn-1*; *R09F10.3*; *sfrp-1*; *Y75B8A.8*) | this paper | | CRISPR RNA guides. Sequences provided in Suppl. File 10 |
| Sequence-based reagent | ssDORT (for *cdk-8*; *gcn-1*; *R09F10.3*; *sfrp-1*) | this paper | | Single-stranded DNA oligonucleotide repair templates. Sequences provided in Suppl. File 10 |
| Peptide, recombinant protein | *Streptococcus pyogenes* Cas9 nuclease V3 | IDT | Cat#:1081058 | |
| Software, algorithm | GATK | DOI: 10.1002/0471250953.bi1110s43 | RRID:SCR_001876 | v3.6 or v3.7 |
| Software, algorithm | breakdancer | DOI:10.1038/nmeth.1363 | RRID:SCR_001799 | v1.4.5-unstable-66-4e44b43 |
| Software, algorithm | pindel | DOI:10.1093/bioinformatics/btp394 | RRID:SCR_000560 | v0.2.5b9, 20160729 |
| Software, algorithm | Tablet | DOI:10.1093/bib/bbs012 | RRID:SCR_000017 | v1.17.08.17 |

*Continued*

| Reagent type (species) or resource | Designation | Source or reference | Identifiers | Additional information |
|---|---|---|---|---|
| Software, algorithm | samtools | DOI:10.1093/bioinformatics/btp352 | RRID:SCR_002105 | 1.9 |
| Software, algorithm | bwa | PMID:19451168 | RRID:SCR_010910 | 0.7.12-r1044 or later |
| Software, algorithm | picard | http://broadinstitute.github.io/picard/ | RRID:SCR_006525 | 1.110 or later |
| Software, algorithm | snpEff | PMID:22728672 | RRID:SCR_005191 | 4.1 g up to 4.3 t |
| Software, algorithm | R project for statistical computing | R Core Team | RRID:SCR_001905 | v3.4.4 |
| Software, algorithm | R package ggplot2 | H. Wickham | RRID:SCR_014601 | v3.2.1 |
| Software, algorithm | R package gridExtra | Baptiste Auguie (2017) | https://CRAN.R-project.org/package=gridExtra | v2.3 |
| Software, algorithm | R package igraph | Csardi G, Nepusz T | https://igraph.org/r/ | v1.1.1 |
| Software, algorithm | R package stats | *R Development Core Team, 2015* | https://www.R-project.org/ | v3.4.4 |
| Software, algorithm | R package fmsb | Minato Nakazawa (2019) | https://CRAN.R-project.org/package=fmsb | v0.6.3 |
| Software, algorithm | R package plyr | Hadley Wickham (2011) | 10.18637/jss.v040.i01 | v1.8.4 |
| Software, algorithm | R package reshape2 | Hadley Wickham (2007) | 10.18637/jss.v021.i12 | v1.4.3 |
| Software, algorithm | R package GenomicFeatures | DOI:10.1371/journal.pcbi.1003118 | RRID:SCR_016960 | v1.30.3 |
| Software, algorithm | R package rtracklayer | DOI:10.1093/bioinformatics/btp328 | http://www.bioconductor.org/ | v1.38.3 |
| Software, algorithm | R Studio Desktop | RStudio Team (2020) | | Version 1.0.143 |

## Nematode strains and culture

All strains used in this study are listed in *Supplementary file 8* with their genotype and origin. MA lines derived from four ancestors and the ancestor stocks were originally obtained from Dr. Charles Baer (*C. elegans* N2 and PB306 and *C. briggsae* HK104 and PB800) (*Baer et al., 2005*). We used MA lines perpetuated by single-hermaphrodite transfer for 250 generations. All lines were cryo-preserved using standard methods (*Stiernagle, 2006*) and freshly thawed prior to experiments.

Unless otherwise stated, all experiments were carried out with strains cultured at 20°C on NGM (Nematode Growth Agar) plates seeded with *Escherichia coli* OP50, following standard procedures (*Brenner, 1974*; *Stiernagle, 2006*).

## Scoring the cell fates of P(3-8).p

Fresh cultures of ancestor and MA lines were regularly thawed from cryopreserved stocks to avoid further drift. All strains were cleaned by hypochlorite treatment (*Stiernagle, 2006*) before initiating experiments. To synchronize nematodes, three to five L4-stage hermaphrodite larvae were transferred to a fresh culture plate at 20°C. When most of their offspring reached the L4 stage (typically after three days, and up to five days for slow-growing strains), vulval cell fates were scored on larvae in the early to mid L4 larval stages, when Pn.p descendants display arrangements typical of each fate. Nematodes were anaesthetized with 1 mM sodium azide and mounted onto an agar pad for Nomarski microscopy observation (*Wood, 1988*). A fusion of P3.p at the L2 stage leaves a single nucleus in the large ventral syncytium ('S' or 4° fate), indicating that P3.p cell exited the vulva differentiation process (*Figure 1A*). The absence of L2 fusion allows P3.p to undergo a round of cell division in the L3 stage, revealed by the presence of two nuclei in the syncytium ('SS' or 3° fate), because its daughter cells also fuse with the syncytium during L3 stage. More rarely, unfused P3.p cells can be partially or fully induced to other vulva fates (2° or 1° fates). The division frequency of P3.p for a line (a binary trait) was estimated on samples of at least 50 nematodes per biological replicate. The number of animals scored per line was a compromise with the number of lines assayed and the number of biological replicates on different days. We use biological replicates in the sense

that the measure was performed on different generations of animals of the same line, assayed on different days. Since P3.p cell fate has been shown to be sensitive to environmental variation (*Braendle and Félix, 2008*), experiments were generally performed by batch including several strains and a common control, for example the ancestral line or the parental line in the case of back-crosses (see below). Masking of the strain identifier was not used. All scores of P3.p division frequency used in this study are provided as *Supplementary file 1*.

## Genomic DNA extraction, library preparation and next-generation sequencing

Whole genomes of six MA lines of interest and their corresponding ancestral strain were re-sequenced. Each strain was freshly thawed and bleached from cryopreserved stocks. The strain was amplified on four 90 mm diameter plates of NEA medium (NGM enriched with agarose [*Richaud et al., 2018*]) seeded with *E. coli* OP50, until the onset of starvation. Nematodes were collected, washed in M9 (*Stiernagle, 2006*) to remove *E. coli*, and centrifuged. A pellet of 200–400 µl of animals was resuspended in 400 µl Cell Lysis Solution (Qiagen Gentra Puregen Cell kit) with 5 µl proteinase K (20 mg/ml) and lysed overnight at 56°C under shaking in Cell Lysis Solution (Qiagen Gentra Puregen Cell kit) with proteinase K (20 mg/ml). Lysates were incubated for 1 hr at 37°C after adding 10 µl of RNAse A (20 mg/ml) and proteins were precipitated with 200 µl of Protein Precipitation Solution (Qiagen Gentra Puregen Cell kit). After centrifugation, DNA was precipitated from the supernatant with 600 µl of isopropanol, washed twice with ethanol 70%, dried for 1 hr and finally resuspended in 100 µl TE buffer. This procedure typically yielded concentrations of ~500 ng/µL (range: 200 ng to 1 µg per µl) of high-quality genomic DNA. Short insert libraries (mean insert size around 500 bp) were prepared by BGI (http://www.genomics.cn/en/index) and paired-end sequenced on Illumina Hiseq2000 with 100 bp reads to obtain 2.2 Gb (aiming at ~20 x mean coverage) of clean data per samples after manufacturer's data filtering (removing adapter sequences, contamination and low-quality reads). Raw sequencing data generated for this study are accessible via the ENA website (https://www.ebi.ac.uk/ena) with accession numbers listed in *Supplementary file 2*.

## Short variant discovery (SNP and short indels)

To efficiently genotype de novo mutations in MA lines and all backcrossed lines, we optimized a procedure of variant discovery with high specificity, avoiding time-consuming assays of false positive calls (*Figure 3—figure supplement 1*). After routine quality checks with FastQC (*Andrews, 2017*), clean reads were mapped using bwa with 'mem' algorithm and '-aM' options (*Li and Durbin, 2009*) to the relevant reference assembly corresponding to WormBase releases WS243 and WS238 for *C. elegans* and *C. briggsae*, respectively (http://www.wormbase.org/). Resulting bam files were further processed with *samtools* (*Li et al., 2009*) to remove unmapped reads or secondary alignments and to keep only mapped reads in a proper pair. The analysis was further performed using the GATK tool suite (*McKenna et al., 2010*) (v3.6 or later) with default parameters (unless otherwise stated), and by adapting the authors' recommendations of best practices (*DePristo et al., 2011*; *Van der Auwera et al., 2013*). Read mappings were pre-processed by tagging duplicate reads with Picard (http://broadinstitute.github.io/picard), by re-aligning reads around indels (GATK tool suite) and by one round of Base Quality Score Recalibration (GATK tool suite) with the HaplotypeCaller tool, resulting in analysis-ready bam files for each sequenced sample. To call short variants (SNPs and indels generally less than 100 bp), these bam files were separately pre-called for variants using the tool HaplotypeCaller in a gVCF mode (option '-ERC GVCF'). Finally, a joint genotyping (with GATK's tool GenotypeGVCFs) was performed using as inputs all the gVCF records of a cohort consisting of the ancestor strain and its derived MA lines. This yielded one unique vcf file per cohort containing the genotypes of all strains of that cohort at each site where at least one strain bears a variation (compared to the reference genome used). We then applied conservative criteria to specifically identify de novo mutations that appeared and fixed during the course of the 250 generations of mutation accumulation. Since all strains are expected to be nearly fully homozygous by constant inbreeding, all heterozygous positions were filtered out. We also removed positions not supported by a coverage superior or equal to 3. Most of the remaining variations are background variations present in the ancestor strain compared to the reference genome of each species, that of strain N2 for *C.*

*elegans* and of strain AF16 for *C. briggsae*. Within a cohort, especially with many MA lines, the variations shared by all strains are very likely ancestral alleles inherited from the ancestor. This high similarity of variation within a cohort was used to increase the specificity of the calls for the PB306 and HK104-derived cohorts (the N2-derived cohort had few background mutations). In both cohorts, background variations were used to train a Variant Quality Score Recalibration (VQSR). In practice, shared variant sites within a cohort were split into background SNPs and indels to perform parallel recalibrations (tool VariantRecalibrator in mode SNP and INDEL, respectively). These training sets of variants were considered to be representative of true sites and were then used to train the model with a prior likelihood of Q12 (93.69%), corresponding to options 'training = true, truth = true, prior = 12.0'. In the case of HK104, we added another training dataset, consisting of validated SNP markers previously used to genotype recombinant progeny between the HK104 and the reference strain AF16 (*Koboldt et al., 2010*; *Ross et al., 2011*) and the 13 new polymorphisms (SNPs or small indels) that were directly validated by pyrosequencing (see below). This additional set is small (948 variants, see the list in vcf format in *Supplementary file 9*) but has a high degree of confidence: we fixed the prior likelihood to Q15 (96.84%) (other parameters of VariantRecalibrator: training = true, truth = true, prior = 12.0). Then, each type of variant was recalibrated (tool ApplyRecalibration) so that 99% of the training dataset should be contained in this quality tranche (option '--ts_filter_level 99.0'). Finally, for each MA line, sites containing an allele passing the VQSR threshold but different from the ancestral line were selected and classified based on the number of other MA lines within the cohort that shared the same genotype. Since spontaneous mutations are rare events and each MA line is an independent replicate of the mutation accumulation experiment, only the variants unique to one MA line were considered as trustful candidates for de novo mutations. Identical mutations found in several MA lines of the same cohort could be either false positive (i.e a background variation present in the ancestor strain that was missed) or a potential mutational hotspot (*Denver et al., 2012*). However, the small size of our cohort does not allow to answer this point. For MA line 516, we simply selected all variants that differed from the re-sequenced N2 ancestor without performing VQSR.

Since repetitive sequences are prone to sequencing or mapping errors, we used versions of reference genomes with masked repetitions, as identified by RepeatMasker software (http://www.repeat-masker.org/) run with default parameters (masked versions are directly available on WormBase, masking 21.9% and 14.6% of bases in *C. elegans* and *C. briggsae* genomes, respectively). However, we observed variations specifically called when using such genome versions, suggesting masking artefacts. To eliminate these, the entire variant discovery pipeline was also applied on the non-masked version of the reference genome and only variations called in both analyses were kept.

## Structural variant discovery

The above procedure only retrieves SNPs and short indels (the longest indel of our final list is 87 bp long, absolute mean indel size is about 18 bp). To detect larger structural variations (SV) like long indels (>100 bp), copy number variations (CNV), repeats (inverted or tandem) or translocations, we used a second approach based on two different complementary callers (*Lin et al., 2015*): the read-pair algorithm Breakdancer (*Chen et al., 2009*) and the split-read algorithm Pindel (*Ye et al., 2009*; *Figure 3—figure supplement 1*). Here again, the whole procedure was optimized to achieve high specificity and reduce false-positive calls. A non-masked version of the genome was used with both programs to generate bam files. In the BreakDancer pipeline, bam files were also filtered to keep only properly mapped reads (see above) and submitted to breakdancer-max command with default options. For each cohort, bam files of the ancestral line and derived MA line(s) were processed in parallel and results were converted to vcf format. For each MAL, variants found in the ancestor line were substracted with leniant criteria to account for the low precision of breakpoint positions achieved by structural variant (SV) callers: two SVs were considered identical if they were of the same type within a 100 bp window (corresponding to read size) and with a difference in size lower than 50%. Then, the following heuristic hard filters were applied (determined on the distribution of the corresponding parameters): QUAL > 90; 50 bp <= SVLEN <= 1 000 000 bp and 25 <= DP <= 150 or 2 <= DP <= 150 for *C. elegans* and *C. briggsae*, respectively. We observed that many false positive calls were generated close to repeated regions where many reads map wrongly. Hence, all variations called in a two kbp region (four times the insert size) where the mean coverage was superior to 100 (five times the mean coverage) were filtered out. Finally, as for short

variations, all MALs of a cohort were compared to keep only unique variations per MAL (using afore-mentioned leniant criteria for SV comparison). In parallel, unfiltered bams were processed with Pin-del (with parameter –max_range_index 6) and for each MAL, variants found in the ancestral line and other MALs of the cohort were filtered out. Finally, the lists of variants generated by Break-dancer and Pindel were intersected to keep only SV called by both procedures. This yielded few can-didate SV (16 for the 6 MALs), all deletions, which were directly inspected with an alignment viewer in both MAL and PL. Only four large deletions passed this ultimate filter. All variants found by the two procedures (short and long variants) are listed in *Supplementary file 3*.

## Variant confirmation and genotyping using pyrosequencing

About 11% of the candidate calls from our short-variant-discovery pipeline were directly tested by pyrosequencing. Variations were not randomly chosen, but selected to be used as helpful genotyp-ing markers during the genetic mapping of the causative locus affecting P3.p division frequency. However, this selection was constrained by the low number of variations per MA line (typically eight per chromosome in *C. elegans* and 34 in *C. briggsae*). Prior to any evidence, two to three variations were selected on each chromosome (ideally one variation in the middle of each chromosomal arm, one in the centromeric region if variations in the arms were excessively shifted to the tips). After the mapping gave the first genetic evidence, additional candidate variations were tested to restrict the mapping interval in the relevant chromosome. SNPs were preferred over indels. Regions containing long stretches of a single nucleotide were avoided, both because the initial call is less likely and because the interpretation of pyrosequencing results is harder in such contexts. Pyrosequencing assays were performed as previously described on a PyroMark Q96 ID instrument (*Besnard et al., 2017*), using universal biotinylated primers (*Duveau and Félix, 2012*). Genotyping assays included the reference genome, the ancestral line and the tested MA line, ie: for the N2 cohort, N2 (reference and ancestor) and MA516; for PB306's cohort, N2 (reference), PB306 (ancestor), and either MA line 418, 450 or 488; for HK104's cohort, AF16 (reference), HK104 (ancestor) and MA296. Candidate SV calls were assayed by PCR with oligonucleotides flanking the predicted deletions. PCR products were controlled on electrophoresis and Sanger-sequenced. Genotyping primers are listed in *Supplementary file 10*.

## Back-crossing MA lines to ancestral line's genetic background

From the initial MA line panel, only MA line 211 was not back-crossed due to time constraints. For all back-crosses, males of the ancestral line were placed with (preferably old, sperm-depleted) her-maphrodites of the mutation accumulation line to back-cross (P$_0$). F1 cross-progeny were isolated on fresh plates and allowed to lay eggs. They were transferred every day to new plates to ease the sep-aration of parents and offspring and synchronization of the F2 offspring. Occasionally, F1 hermaph-rodites were eventually lysed and genotyped by pyrosequencing to ensure they were true cross progeny. Several F2 animals were isolated for each cross and gave rise to an independent line of one back-cross increment compared to the initial P0. Serial back-crosses are noted as 1X, 2X, etc. Different strategies and crossing schemes were applied for the different MA lines (*Figure 3—figure supplements 4–8*). The first strategy consisting in crossing without selection was applied for the first back-cross of MAL296 and the second back-cross of MAL516. In this case, several random F2 her-maphrodites were isolated, without scoring the vulva or selecting for any other phenotype. A second strategy consisted in selecting F2 based on a phenotype. MAL296 2X and 3X lines were generated by selecting F2 hermaphrodites showing a divided ('SS') fate for P3.p. MAL516 1X lines were gener-ated by selecting for Egl (egg-laying) or Pvl (protruding vulva) phenotypes, which were apparent in MAL516. Back-crossed lines of the PB306 cohort (MAL 418, 450 and 488) were generated by select-ing a Mendelian recessive (dumpy, small, slow-growth or low-brood-size) phenotype versus wild-type F2 hermaphrodites, in equal amounts. Indeed, all three parent MA lines present a mixture of these phenotypes: this strategy was designed to test a linkage between these obvious morphologi-cal phenotypes and P3.p cell fate. Since the linkage was confirmed at each back-cross level, these selection criteria were kept over serial back-crosses (up to 4X for MAL 418 and 450). For these two lines, the morphological phenotype was used to accelerate the crossing scheme: wild-type F1 her-maphrodites (necessarily cross-progeny given the recessive transmission of morphological defects) were directly crossed with PB306 males, resulting in new F1 progeny that were isolated on fresh

plates. Due to Mendelian segregation, only half of these new F1 carried a mutant allele and segregates mutant F2 progeny. Only these F1 plates were retained to select both mutant and WT F2. Resulting lines have two increment back-cross levels compared to the initial P0 (for instance, 4X starting from a 2X-line).

In all strategies, F2s were singled on fresh plates and perpetuated in parallel by single-hermaphrodite transfer for four to five generations to maximize homozygosity at all loci, and finally amplified for cryo-preservation.

## Mapping the causal mutation

For each MA line, the set of validated de novo mutations constituted genetic markers spanning all chromosomes. Independent back-crossed lines were scored for P3.p behaviour and then genotyped for some of these markers in order to identify first a linked chromosome, and then a shorter interval depending on the availability of markers (See *Figure 3* and *Supplementary file 4*). All lines were not systematically genotyped for all markers, except for the candidate mutation.

## CRISPR/Cas9 genome editing

CRISPR/Cas9-mediated homologous recombination (HR) was used to mimic the candidate mutation of MA lines 296, 450, 488 and 516. HR was performed using single-stranded DNA oligonucleotide repair templates (ssDORT) with 35 bp 5' and 3' homology arms, following a combination of previously described methods (*Paix et al., 2017b*; *Paix et al., 2017a*; *Dokshin et al., 2018*). Briefly, the trans-activating CRISPR RNAs (tracrRNAs; ordered from IDT) were individually annealed with CRISPR RNA guides (crRNAs) by incubation at 95°C for 5 min and cooling to room temperature (~23–25°C) for another 5 min to generate single-guide RNAs (sgRNA). Then, recombinant *Streptococcus pyogenes* Cas9 nuclease V3 (IDT) was incubated with sgRNAs for 10 min at 37°C to form ribonucleoprotein complexes. Next, ssDORTs, plasmids and nuclease-free water were added to the mix and centrifuged at 10,000 rpm for 2 min before loading into the needle. The mixes were micro-injected into gonads of 1 day old adult hermaphrodites (P0) of the ancestral lines. F1 progeny was singled from plates displaying the highest number of dumpy (Dpy) or roller (Rol) phenotypes. Two days later, single F1s were PCR screened for HR replacements using primers flanking the target region (outside the ssDORT sequence) and one HR-specific primer. Non-Rol or non-Dpy progeny (F2 or F3) of positive F1 animals were singly propagated to generate homozygous progeny and further genotyped by PCR. Genomic replacements were confirmed by Sanger sequencing. crRNAs were designed in http://crispr.mit.edu/ (Zhang lab) for *C. elegans* editings and http://crispor.tefor.net/ for *C. briggsae*, and ordered from IDT.

To generate the large deletion of 1382 bp in the exon 21 of *gcn-1* (as found in MAL516), we used two crRNAs (crRNA.gcn-1.E21.prox.g1 and crRNA.gcn-1.E21.dist.g1) to generate double strand breaks (DSB) flanking the deletion breakpoints and a ssDORT (gcn-1.E21.rt) to generate the large deletion by HR repair. We used the following injection mix: 0.25 µg/µl Cas9 protein (IDT), 57 µM tracrRNA, 22.5 µM of crRNA.gcn-1.E21.prox.g1 and 22.5 µM of crRNA.gcn-1.E21.dist.g1, 110 ng/µl gcn-1.E21.rt4 repair template, 40 ng/µl of the plasmid pRF4::*rol-6(su1006)* as an injection marker, and 50 ng/µl of empty pBluescript plasmid. The mix was injected into gonads of 1 day old adult N2 hermaphrodites (Baer 'ancestral N2' stock).

The missense mutation in codon 40 from a valine (C$\underline{T}$T) into an alanine (G$\underline{C}$T) of *cdk-8* (as found in MAL450) was generated using a crRNA guide (crRNA.cdk-8.E2.g1) that induces a DSB located 11 bp from the target region and a ssDORT (cdk-8.E2.rt1) with the missense mutation and nine silent mutations to prevent Cas9 re-cutting and minimise template switching. To control for the silent mutations, we generated control lines with another ssDORT (cdk-8.E2.rt2) that only has the nine silent mutations. We used the following injection mix: 0.3 µg/µl Cas9 protein (IDT), 40 µM tracrRNA and 30 µM of crRNA.cdk-8.E2.g1, 10 µM tracrRNA and 7.5 µM of crRNA.dpy-10 (IDT) as a co-CRISPR marker, 110 ng/µl cdk-8.E2.rt1 repair template (or cdk-8.E2.rt2), 50 ng/µl of empty pBluescript plasmid, and 0.5 µM *dpy-10* repair template. The mix was injected into gonads of 1-day-old adult ancestral PB306 hermaphrodites.

To generate the 16 bp deletion in the exon 4 of the *R09F10.3* locus (as found in MAL488), we used a crRNA guide (crRNA.R09F10.3.E4.g1) to generate a DSB in the target region and a ssDORT (R09F10.3_E4.rt1) to generate the small deletion by HR repair using the following injection mix: 0.3

µg/µl Cas9 protein (IDT), 40 µM tracrRNA and 30 µM of crRNA.R09F10.3.E4.g1, 10 µM tracrRNA and 7.5 µM of crRNA.dpy-10 (IDT) as a co-CRISPR marker, 110 ng/µl R09F10.3_E4.rt1 repair template, 50 ng/µl of empty pBluescript plasmid, and 0.5 µM *dpy-10* repair template. The mix was injected into gonads of 1 day old adult ancestral PB306 hermaphrodites.

The missense mutation in codon 59 from an asparagine (<u>A</u>AT) to a histidine (<u>C</u>AT) of *sfrp-1* in *C. briggsae* (as found in MAL296) was edited using a crRNA guide (crRNA.sfrp-1.E2.g1) that induces a DSB 10 bp from the target region and a ssDORT (sfrp-1.E2.rt1) with the missense mutation and eight silent mutations. To control for the eight silent mutations, we generated control lines with another ssDORT (sfrp-1.E2.rt2) that only has the silent mutations. We used the following injection mix: 1 µg/µl Cas9 protein (IDT), 30 mM KCl and 4 mM HEPES pH7.5, 40 µM tracrRNA and 30 µM of crRNA. sfrp-1.E2.g1, 10 µM tracrRNA and 7.5 µM of crRNA.dpy-1 as a co-CRISPR marker, 110 ng/µl sfrp-1. E2.rt1 repair template (or sfrp-1.E2.rt2), and 50 ng/µl of empty pBluescript plasmid. The mix was injected into gonads of 1-day-old adult ancestral HK104 hermaphrodites.

To validate the 54,355 bp deletion on the chromosome IIIR of MA line 418 and identify the causal gene(s), we generated frameshifting indels in the seven protein-coding genes within the deleted region, using CRISPR/Cas9 editing without repair template (non-homologous end-joining) as described in *Friedland et al., 2013*; *Arribere et al., 2014*. Guide RNAs were designed with the CRISPOR online program (*Haeussler et al., 2016*). To generate the pU6-*target*-sgRNA plasmid, we replaced the *dpy-10* target site with the desired target gene site in the pJA58 plasmid (*Arribere et al., 2014*), using the Q5 Site-Directed Mutagenesis Kit (New England BioLabs) and the online tool NEBasechanger to design the mutagenesis primers. For genome editing, young adult PB306 hermaphrodites were injected with the following injection mix: 100 ng/µl of the pU6-target-sgRNA plasmid, 50 ng/µl of *Peft-3::Cas9-SV40NLS::tbb-2 3'UTR* plasmid (*Friedland et al., 2013*), 60 ng/µl pJA58 plasmid as co-CRISPR marker and 10 ng/µl of the pPD118.33 plasmid (*Pmyo2::GFP*) as co-injection marker. We then singled the F1 progeny from plates with a high number of animals displaying the Dpy phenotype and GFP expression. F1s were screened by PCR for indels with flanking primers. Non-Dpy progeny of positive F1s were rendered homozygous and mutations were characterized by Sanger sequencing.

All oligonucleotides used for CRISPR/Cas9-mediating genome editing (guides, repair templates and genotyping primers) are listed in *Supplementary file 11*. The sequences in ancestor line, MA line and edited lines are provided in *Supplementary file 5*.

## Genomic analysis and data visualization

The GC content of DNA sequences was computed using *bedtools* (*Quinlan, 2014*). Extraction of sequences with different annotations was performed using the R package 'GenomicFeatures' (*R Development Core Team, 2015*; *Lawrence et al., 2013*). Repeats were retrieved from 'masked' genome fasta files available from Wormbase (WS243 and WS234 for *C. elegans*; WS238 for *C. briggsae*). To compare mutation rates in other *C. elegans* datasets, the mutation found in *C. briggsae sfrp-1* was transposed to the homologous base pair in *C. elegans*. Additional filters were applied to the published list of mutations found in the dataset of *Saxena et al., 2019*, in order to remove most likely false positive calls: overlapping SNPs or indels at the same locus in the same line (initial calling procedure was performed separately), SNPs at 2 bp or less from an indel in the same line, identical mutations shared by related lines (likely background mutations), groups of identical mutations over large chromosomal regions found in multiple lines (possible cross-contamination during sequencing). Since this previous study did not look for large structural variants, we systematically looked with the Tablet alignment viewer (*Milne et al., 2013*), using bam files kindly provided by the authors, for large deletions falling in the exons of the five causal genes in all the MA lines of the dataset and tested all dubious instances by direct PCR and Sanger sequencing (see *Supplementary file 7b*) for the list of tested MA lines and re-sequenced genomic regions (corresponding PCR oligonucleotide sequences are listed in *Supplementary file 10b*). We did not detect any structural variants in the exons of the five causal genes. Functional annotations of natural polymorphisms were predicted using snpEff. How snpEff classifies the putative effect of genomic variants into high or moderate is available online (https://www.elegansvariation.org/help/Variant-Prediction/). Computing and plotting different genomic features (*Figures 5* and *6*) was performed with R using custom scripts and the ggplot2 package.

## Statistical analysis

Differences between P3.p division frequencies were evaluated using pair-wise Fisher exact tests with false-discovery rate (fdr) level of 0.05 to correct for multiple testing (R, fmsb package). The resulting pair-wise matrix of adjusted $p$-values was used to generate post-hoc labeling of each strain. Other statistics were computed using R, stats package (*R Development Core Team, 2015*) (specifically confidence intervals with prop.test, Pearson's correlation test with cor. test and $X^2$-test with chisq. test).

All raw sequencing data supporting the conclusions of this article have been submitted to ENA. Study and sample accession numbers corresponding to the sequencing data of the ancestor and MA lines used in this study are listed in *Supplementary file 2*. Custom scripts were used to pipeline the different tools during the variant analysis (bash/python scripts) or to perform statistical analysis and to plot results (R scripts).

The data set containing all mutations in the mutagenized strains of the Million Mutation Project was downloaded online (http://genome.sfu.ca/mmp/mmp_mut_strains_data_Mar14.txt). Hard-filtered Variant data of the latest release of the Caenorhabditis Natural Diversity Resource (release ID: 20180527) was downloaded online (https://www.elegansvariation.org/data/release/latest).

## Acknowledgements

We are very grateful to Ayush Saxena, Michael Snyder and Charles Baer for sharing data, strains and DNA. Some strains were provided by the *Caenorhabditis* Genetics Center, which is funded by the National Institutes of Health Office of Research Infrastructure Programs (P40 OD-010440). We acknowledge WormBase, CeNDR, the Million Mutation Project and the National Bioresource Project of the Mitani laboratory. We gratefully acknowledge the PSMN (Pôle Scientifique de Modélisation Numérique) computing center of ENS-Lyon for support during the latter part of the genomic analysis. The authors declare that they have no competing interests.

## Additional information

### Funding

| Funder | Grant reference number | Author |
| --- | --- | --- |
| Agence Nationale de la Recherche | ANR-12-BSV2-0004-01 | Marie-Anne Félix |
| Agence Nationale de la Recherche | ANR-18-CE13-0006-01 | Marie-Anne Félix |
| H2020 Marie Skłodowska-Curie Actions | Training Grant 751530-EvoCellFate | Joao Picao-Osorio |

The funders had no role in study design, data collection and interpretation, or the decision to submit the work for publication.

### Author contributions

Fabrice Besnard, Formal analysis, Investigation, Visualization, Methodology, Writing - original draft; Joao Picao-Osorio, Formal analysis, Funding acquisition, Validation, Investigation, Writing - review and editing; Clément Dubois, Investigation, Writing - review and editing; Marie-Anne Félix, Conceptualization, Supervision, Funding acquisition, Writing - original draft

### Author ORCIDs

Fabrice Besnard (iD) https://orcid.org/0000-0003-4619-5547

### Decision letter and Author response

Decision letter https://doi.org/10.7554/eLife.54928.sa1
Author response https://doi.org/10.7554/eLife.54928.sa2

# Additional files

## Supplementary files

- Supplementary file 1. P3.p division frequency scoring. n (last column) is the number of animals.

- Supplementary file 2. Accession numbers for the sequencing data.

- Supplementary file 3. Mutations found in MA lines. The columns are named according to the vcf format. In addition, column I provides the identifier of the tested marker and column H whether it was validated. The lines highlighted with a red background are the causative mutations.

- Supplementary file 4. Genetic mapping of causative mutations in MA lines. The first sheet provides the summary of the interval. Each successive sheet shows the backcross genotyping and phenotying (column F gives the statistical groups computed in *Figure 3*): 'AL' as in the ancestor line; 'MAL' as in the Mutation Accumulation line; ND: not determined; HET: heterozygote.

- Supplementary file 5. Sequences of the causal mutations in the MA lines and of the CRISPR edits.

- Supplementary file 6. List of pleiotropic phenotypes observed in selected Mutation Accumulation Lines and CRISPR genome editings.

- Supplementary file 7. Analysis of mutations found around the five causal mutations in the three comparison datasets. (a) No mutations were found in the vicinity of the five causal mutations in the MA line dataset, and the regions do not contain a particularly high level of mutations/polymorphisms in the MMP and CeNDR datasets. (b) The second sheet provides the list of mutation accumulation lines from *Saxena et al., 2019*.

- Supplementary file 8. Strains used in this study. The superscript 'CB' (e.g. N2[CB]) refers to the strain origin in Charles Baer's laboratory.

- Supplementary file 9. List of high-confidence variants between the *C. briggsae* strains HK104 and AF16. File is in vcf format. This list was used as prior knowledge for the VQSR procedure computed with GATK (see Materials and methods).

- Supplementary file 10. Genotyping primers for mutation accumulation lines. (a) Genotyping of MA lines from this study. (b) Re-sequencing of MA lines from *Saxena et al., 2019*.

- Supplementary file 11. Oligonucleotides used for CRISPR/Cas9 genome edition.

- Transparent reporting form

## Data availability

Sequencing data have been deposited at EBI under accessions PRJEB30820-2. All other data generated or analysed during this study are included in the manuscript and supporting files. Source data files have been provided in Supplementary File 1.

The following datasets were generated:

| Author(s) | Year | Dataset title | Dataset URL | Database and Identifier |
|---|---|---|---|---|
| Besnard F, Félix M-A | 2019 | Whole-genome re-sequencing of the wild accession HK104 (*Caenorhabditis* briggsae nematode) and two derived Mutation Accumulation Lines | http://www.ebi.ac.uk/ena/data/view/PRJEB30820 | EBI, PRJEB30820 |
| Besnard F, Félix M-A | 2019 | Whole-genome re-sequencing of the reference strain N2 (*Caenorhabditis elegans* nematode) and one derived Mutation Accumulation Line. | http://www.ebi.ac.uk/ena/data/view/PRJEB30821 | EBI, PRJEB30821 |
| Besnard F, Félix M-A | 2019 | Whole-genome re-sequencing of the wild isolate PB306 (*Caenorhabditis elegans* nematode) and three derived Mutation Accumulation Lines | http://www.ebi.ac.uk/ena/data/view/PRJEB30822 | EBI, PRJEB30822 |

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
