## [Decision Letter]

**Acceptance summary:**

One of the most intriguing observations in ecology and evolution relates to the large breadth of rates at which traits evolve. Whether this variation is caused by natural selection, variation in mutation rates or by differences in the organization of the underlying molecular network is largely unexplored. Here, the authors examine whether the rapid evolution of a developmental trait in worms is caused by its underlying architecture, more specifically by the size of its mutational target, or by a high mutation rate of the genes involved in shaping this trait. They find that the rapid evolution of this trait is not caused by a high mutation rate of the underlying genes, but rather by the large number genes involved. These results show that how traits are organized at the molecular level influences their rate of evolution, demonstrating how important it is to integrate cell biology and quantitative genetics to fully understand evolution.

**Decision letter after peer review:**

Thank you for submitting your article "A broad mutational target explains a fast rate of phenotypic evolution" for consideration by *eLife*. Your article has been reviewed by three peer reviewers, one of whom is a member of our Board of Reviewing Editors, and the evaluation has been overseen by Detlef Weigel as the Senior Editor. The following individual involved in review of your submission has agreed to reveal their identity: David Matus (Reviewer #2).

The reviewers have discussed the reviews with one another and the Reviewing Editor has drafted this decision to help you prepare a revised submission.

Summary:

The authors identified developmental traits in worms that evolve particularly rapidly in MA lines. Two factors could explain this rapid evolution.

1) A large number of loci involved in determining or affecting the trait.

2) A few loci but a high rate of mutation.

These two potential factors underlying high rate of evolution have rarely been disentangled because they require that one identifies the mutations underlying variation in the trait. Here the authors do so using the nematode vulva precursor cells and find that the high mutational variance (Vm) for this trait is likely elevated due to the large underlying mutational target size.

The reviewers agree that the study is well performed, that the question is of great interest and that the manuscript is well written and presented. The comments to be addressed are merged below into a single list.

Essential revisions:

A lot of emphasis is given to highly mutable sites such as in short tandem repeats. These are highly mutable, but they may at the same time be underrepresented in coding and functional parts of the genome. The authors eliminate the possibility of high mutation rates by not observing causal mutations in the repeats. If these repeats are rare in functional parts of the genome, the contribution of mutation rate to high Vm may come from other types of more mutable sites and not these extreme cases. Since mutation rate in *C. elegans* has been measured using MA lines and there is a lot of data on population variation, it would be useful if the author could also examine the relative expected rates for the different types of mutations (and flanking context) they observe as causal. They could see at least in terms of ranking, whether these changes are more or less common. The genes involved for instance could be in genomic regions with high or low GC content, which could have high or low mutation rates, etc. Overall, eliminating the role of mutation rate is done in a qualitative manner and it could be much more quantitative and better reflect the continuous distribution of mutation rates and mutational target sites. I understand that this is difficult to do with a small set of mutations but if the authors could at least verify that the types of mutation they see do not exceptionally occur at high frequency that would better support their conclusion.

Another comment would be about the focus on developmental systems to study the underlying factors of variation in Vm. Some significant work has been done for instance on the underlying basis of gene expression variation in model organisms and in parallel in MA lines. Also, attempts have been made at estimating the relationship between Vm and mutational target sites in some cases. It would be good to acknowledge/discuss this work as it would show that these concepts and factors are probably true for any measurable trait and that there may be common rules across types of traits. This would make the paper appeal to a wider audience than the evodevo community.

Another critique or comment is less of a critique and more of a suggestion – there's an opportunity lost in not presenting this work as also an incredibly innovative way to leverage quantitative genetics to identify novel regulators of a biological process. Also, as a cell biologist, I am left with wanting to know whether the genes/pathways identified function autonomously or not. I fully recognize that this may be beyond the scope of the present study, but from a developmental perspective and as a strategy for identifying novel members and pathways that contribute to the evolution and development of a particular biological process, it would be interesting to know whether the newly identified pathways function autonomously to mediate P3.p fusion/division.

One potential limitation to their study is that so few lines were studied. They originally evaluated 15 lines, chose 6 due (presumably) to inconsistent P3.p division frequency results with the previous study, and then removed 1 more due to time constraints. The remaining 5 are split over two species. These lines are not representative in a random sampling sense – it isn't clear to me that we definitively know whether they are representative in terms of the biology and the question they want to answer. However, there are reasons to think that they are – at least with respect to the main question of the paper. They address this somewhat in their Discussion in the paragraph starting "An obvious further question…". I am not advocating that they look at more lines. They discuss the practical difficulties of doing this experiment, there may not be more lines to look at (since they had to exclude a bunch for inconsistency), and even if they haven't definitively rejected the mutagenic DNA hypothesis for all effect sizes of mutations, they have certainly shown that it isn't an explanation for their specific data. There are enough different pathways involved among their 5 mutations that it is far easier to think that there is plenty of opportunity for mutations of small effect to arise in those pathways without postulating that there is an unknown but much more important class of P3.p affecting mutagenic loci that are really behind any smaller evolved differences in others of the MA lines. Indeed, the more likely situation is that the mutational target size is even bigger than they found and that sampling more lines would just reinforce their conclusion.

---

## [Author Response]

Essential revisions:A lot of emphasis is given to highly mutable sites such as in short tandem repeats. These are highly mutable, but they may at the same time be underrepresented in coding and functional parts of the genome. The authors eliminate the possibility of high mutation rates by not observing causal mutations in the repeats. If these repeats are rare in functional parts of the genome, the contribution of mutation rate to high Vm may come from other types of more mutable sites and not these extreme cases. Since mutation rate in *C. elegans* has been measured using MA lines and there is a lot of data on population variation, it would be useful if the author could also examine the relative expected rates for the different types of mutations (and flanking context) they observe as causal. They could see at least in terms of ranking, whether these changes are more or less common. The genes involved for instance could be in genomic regions with high or low GC content, which could have high or low mutation rates, etc. Overall, eliminating the role of mutation rate is done in a qualitative manner and it could be much more quantitative and better reflect the continuous distribution of mutation rates and mutational target sites. I understand that this is difficult to do with a small set of mutations but if the authors could at least verify that the types of mutation they see do not exceptionally occur at high frequency that would better support their conclusion.

The reviewers may have missed that we had analyzed GC content around the mutations and plotted them in a quantitative manner along the genomic distribution (now Figure 5B and Figure 5—figure supplement 1): this analysis did not point to particularly unusual values. We also had provided quantitative data with two measures of the genomic context related to repeats: the distribution of distances to the closest repeat (Figure 5B) and the repeat content at a larger scale (50 kb-window, Figure 5—figure supplement 1), which did not show unusual values.

Although this is a qualitative argument, we had also pointed out the five mutations are so diverse that higher mutability for all these different cases is unlikely: diversity of the type of mutations (Figure 4A and Figure 5A), of their relation to functional sequences (Figure 5A) and of their chromosomal locations (Figure 5A and Figure 5—figure supplement 1).

However, it was useful to add further quantitative analyses of mutation rates. We thus performed new analyses that lead to enrich this revised manuscript with a new main figure (Figure 6), a new supplementary figure (Figure 6—figure supplement 1), a modified supplementary figure (Figure 5—figure supplement 1), a new supplementary table (Supplementary file 7) and substantial text modification corresponding to this part.

Ideally, sequencing thousands of mutation accumulation lines should cover almost all the genome of a *Caenorhabditis* species with spontaneous mutations, so that the "continuous distribution of mutation rates and target sites" could be computed and the five causal loci of the study positioned in this distribution. Unfortunately, such unrealistic data set does not exist, so a fine measure of spontaneous mutation rates is not possible.

However, as suggested by the reviewers, existing data, yet imperfect, allow to compare the mutations identified in this study with mutations obtained in different contexts, providing a more subtle understanding of the range of mutation rates and their likelihood. Notably, we provide data on mutability not only for the five causal nucleotide positions identified (Figure 5 and Figure 5—figure supplement 1), but also for the five causative genes leading to P3.p evolution (Figure 6 and Figure 6—figure supplement 1). To this end, we combined data of mutations from different sources to mitigate their individual shortcomings. Data from other Mutation Accumulation Lines (Saxena et al., 2019) provide the perfect comparison but they are largely under-powered. Data from the Million Mutation Project (Thompson et al., 2013) provide enough power, but induced and spontaneous mutations surely have a different spectrum so that the type and target sites of mutations are not strictly comparable. Natural variation (Cook et al., 2017) is abundant and originates from spontaneous mutations with a supposedly similar spectrum as laboratory MA lines, but their final distribution is modified by the action of natural selection and genetic draft.

Here are listed the main conclusions drawn from these new analyses:

Regarding mutated positions (nucleotides)

– According to statistics of mutation accumulation in *C. elegans* derived from previous studies (Denver et al., 2012 and Saxena et al., 2019), the three small mutations (two SNPs and a 16-bp deletion) are not the most frequent: low frequencies for these single nucleotide substitutions, low frequencies of corresponding 3-bp motifs, small indels are more infrequent than SNPs, mutations (of all type) outside repeats are less frequent (subsection “Molecular nature of the causal mutations and mutation rates at these loci”).

– Parsing whole-genome sequences of 75 other MA lines of *C. elegans* (Saxena et al., 2019) reveals that the causal loci (nucleotide positions of mutations or deletion breakpoints) are not hit again (even neighbouring sequences up to several kb) nor the causative genes (exons), excluding the extreme scenario of super-mutable loci and/or genes (subsection “Molecular nature of the causal mutations and mutation rates at these loci”, Figure 6A).

– Using the MMP and CeNDR, none of the five nucleotide positions (breakpoints for deletions) were particularly prone to mutations (Supplementary file 7).

Regarding mutated genes

The high Vm of P3.p fate is ultimately caused by functional impacts on underlying genes, a first level in genotype/phenotype mapping. In the MMP and CeNDR, the mutation rate of the five causative genes is mainly driven by their size (which correlates with large introns and repeats) (Figure 6B). One of the five genes *(gcn-1*) is particularly long (the 10th longest protein-coding gene in *C. elegans*) so it is particularly likely to be the target of mutations.

We hope that these new elements answered the reviewers' concerns.

Another comment would be about the focus on developmental systems to study the underlying factors of variation in Vm. Some significant work has been done for instance on the underlying basis of gene expression variation in model organisms and in parallel in MA lines. Also, attempts have been made at estimating the relationship between Vm and mutational target sites in some cases. It would be good to acknowledge/discuss this work as it would show that these concepts and factors are probably true for any measurable trait and that there may be common rules across types of traits. This would make the paper appeal to a wider audience than the evodevo community.

We did not mean to emphasize developmental systems particularly and rather had sought to extend the significance of our work to any case of genotype-phenotype mapping. Our title is generic and the Abstract does not mention development either. We use the word "development" in "developmental constraints" because of its historical importance.

We now changed: "the second at the developmental level" by "the second at the level of genotype-phenotype mapping"; "development of such a trait" to "construction of such a trait". We kept the word "development" when referring specifically to vulva development.

Besides generic references that apply to any living organisms with a G/P map, we had provided in the first version examples of relationship between mutational variance and phenotypic variation in a diversity of animals as well as in several articles in *S. cerevisiae*, *Aspergillus* (fungi), and *Arabidopsis thaliana* (plant). We now add further references including in bacteria (Girgis et al., 2009, Girgis et al., 2012, Fridman et al., 2014, Brauner et al., 2016, Khare and Tavazoie, 2020), as well as examples using the multidimensional nature of the transcriptome (Denver et al., 2005, Rifkin et al., 2005, Landry et al., 2007, Hine et al., 2018). We hope this is satisfactory.

Another critique or comment is less of a critique and more of a suggestion – there's an opportunity lost in not presenting this work as also an incredibly innovative way to leverage quantitative genetics to identify novel regulators of a biological process. Also, as a cell biologist, I am left with wanting to know whether the genes/pathways identified function autonomously or not. I fully recognize that this may be beyond the scope of the present study, but from a developmental perspective and as a strategy for identifying novel members and pathways that contribute to the evolution and development of a particular biological process, it would be interesting to know whether the newly identified pathways function autonomously to mediate P3.p fusion/division.

Thank you for pointing to this possible follow-up of our work. We agree but leave this point for future studies. We added in the first paragraph of the Discussion: "Using this quantitative genetics approach, we were able to find new regulators of P3.p developmental fate that are available for further developmental studies."

One potential limitation to their study is that so few lines were studied. They originally evaluated 15 lines, chose 6 due (presumably) to inconsistent P3.p division frequency results with the previous study, and then removed 1 more due to time constraints. The remaining 5 are split over two species. These lines are not representative in a random sampling sense – it isn't clear to me that we definitively know whether they are representative in terms of the biology and the question they want to answer. However, there are reasons to think that they are – at least with respect to the main question of the paper. They address this somewhat in their Discussion in the paragraph starting "An obvious further question…". I am not advocating that they look at more lines. They discuss the practical difficulties of doing this experiment, there may not be more lines to look at (since they had to exclude a bunch for inconsistency), and even if they haven't definitively rejected the mutagenic DNA hypothesis for all effect sizes of mutations, they have certainly shown that it isn't an explanation for their specific data. There are enough different pathways involved among their 5 mutations that it is far easier to think that there is plenty of opportunity for mutations of small effect to arise in those pathways without postulating that there is an unknown but much more important class of P3.p affecting mutagenic loci that are really behind any smaller evolved differences in others of the MA lines. Indeed, the more likely situation is that the mutational target size is even bigger than they found and that sampling more lines would just reinforce their conclusion.

Of course the mutational target size is larger than what we found with these five mutations. We added "This is a small sample of possible mutations and already…".